# Death following traumatic brain injury in *Drosophila* is associated with intestinal barrier dysfunction

Rebeccah J Katzenberger[1], Stanislava Chtarbanova[2], Stacey A Rimkus[1], Julie A Fischer[1], Gulpreet Kaur[1,3], Jocelyn M Seppala[1], Laura C Swanson[1,4], Jocelyn E Zajac[1], Barry Ganetzky[2]*, David A Wassarman[1]*

[1]Department of Cell and Regenerative Biology, School of Medicine and Public Health, University of Wisconsin-Madison, Madison, United States; [2]Laboratory of Genetics, University of Wisconsin-Madison, Madison, United States; [3]Graduate Program in Cellular and Molecular Biology, University of Wisconsin–Madison, Madison, United States; [4]Medical Scientist Training Program, School of Medicine and Public Health, University of Wisconsin-Madison, Madison, United States

**Abstract** Traumatic brain injury (TBI) is a major cause of death and disability worldwide. Unfavorable TBI outcomes result from primary mechanical injuries to the brain and ensuing secondary non-mechanical injuries that are not limited to the brain. Our genome-wide association study of *Drosophila melanogaster* revealed that the probability of death following TBI is associated with single nucleotide polymorphisms in genes involved in tissue barrier function and glucose homeostasis. We found that TBI causes intestinal and blood–brain barrier dysfunction and that intestinal barrier dysfunction is highly correlated with the probability of death. Furthermore, we found that ingestion of glucose after a primary injury increases the probability of death through a secondary injury mechanism that exacerbates intestinal barrier dysfunction. Our results indicate that natural variation in the probability of death following TBI is due in part to genetic differences that affect intestinal barrier dysfunction.

*For correspondence:
ganetzky@wisc.edu (BG);
dawassarman@wisc.edu (DAW)

**Competing interests:** The authors declare that no competing interests exist.

## Introduction

Traumatic brain injury (TBI) is the leading cause of death for people under the age of 44 in the United States (*Harrison-Felix et al., 2009*; *Coronado et al., 2011*). Death following TBI is not only due to primary injuries, that is, mechanical injuries that occur at the moment of impact to the brain, but also to secondary injuries, that is, non-mechanical injuries that evolve over time in response to primary injuries (*Masel and DeWitt, 2010*; *Blennow et al., 2012*; *Prins et al., 2013*). Because secondary injuries are non-mechanical and are delayed relative to primary injuries they may be sensitive to therapeutic interventions. For example, secondary injuries to the intestine can rapidly follow primary injuries to the brain, and interventions that block intestinal injuries may prevent bacterial translocation and subsequent sepsis (*Hang et al., 2003*; *Feighery et al., 2008*; *Jin et al., 2008*; *Bansal et al., 2009*, *2010*). However, present understanding of the cellular and molecular mechanisms that underlie secondary injuries is not yet sufficient to develop therapeutic interventions (*Menon, 2009*; *Xiong et al., 2013*).

We have used a *Drosophila melanogaster* model to investigate the mechanisms underlying secondary injuries that cause death following traumatic injury. Our fly model uses the high-impact trauma (HIT) device, consisting of a metal spring with a stationary end attached to a board and a free end positioned over a polyurethane pad, to inflict traumatic injury (*Katzenberger et al., 2013*). A plastic vial containing unanesthetized flies is connected to the free end. When the spring is

**eLife digest** Traumatic brain injury (TBI) caused by a violent blow to the head or body and the resultant collision of the brain against the skull is a major cause of disability and death in humans. Primary injury to the brain triggers secondary injuries that further damage the brain and other organs, generating many of the detrimental consequences of TBI. However, despite decades of study, the exact nature of these secondary injuries and their origin are poorly understood. A better understanding of secondary injuries should help to develop novel therapies to improve TBI outcomes in affected individuals.

To obtain this information, in 2013 researchers devised a method to inflict TBI in the common fruit fly, *Drosophila melanogaster*, an organism that is readily amenable to detailed genetic and molecular studies. This investigation demonstrated that flies subjected to TBI display many of the same symptoms observed in humans after a brain injury, including temporary loss of mobility and damage to the brain that becomes worse over time. In addition, many of the flies die within 24 hr after brain injury.

Now Katzenberger et al. use this experimental system to investigate the secondary injuries responsible for these deaths. First, genetic variants were identified that confer increased or decreased susceptibility to death after brain injury. Several of the identified genes affect the structural integrity of the intestinal barrier that isolates the contents of the gut—including nutrients and bacteria—from the circulatory system. Katzenberger et al. subsequently found that the breakdown of this barrier after brain injury permits bacteria and glucose to leak out of the intestine. Treating flies with antibiotics did not increase survival, whereas reducing glucose levels in the circulatory system after brain injury did. Thus, Katzenberger et al. conclude that high levels of glucose in the circulatory system, a condition known as hyperglycemia, is a key culprit in death following TBI.

Notably, these results parallel findings in humans, where hyperglycemia is highly predictive of death following TBI. Similarly, individuals with diabetes have a significantly increased risk of death after TBI. These results suggest that the secondary injuries leading to death are the same in flies and humans and that further studies in flies are likely to provide additional new information that will help us understand the complex consequences of TBI. Important challenges remain, including understanding precisely how the brain and intestine communicate, how injury to the brain leads to disruption of the intestinal barrier, and why elevated glucose levels increase mortality after brain injury. Answers to these questions could help pave the way to new therapies for TBI.

deflected and released, the vial rapidly strikes the pad, and a mechanical force is delivered to the flies as they impact the vial wall. A high-speed movie shows that a strike from the HIT device causes flies to hit the vial wall multiple times with their head and body, probably inflicting traumatic injury to multiple organs, including the brain (*Balsiger et al., 2014*). Closed-head TBI may result from impacts to the head or body that cause the fly brain to ricochet and deform against the head capsule, similar to what happens to humans in falls and car crashes (*Davceva et al., 2012*). Accordingly, flies treated with the HIT device display phenotypes consistent with brain injury, including temporary incapacitation followed by ataxia, gradual recovery of mobility, neurodegeneration over time, and death within 24 hr (*Katzenberger et al., 2013*). However, as in polytraumatic injuries in humans (e.g., blast injuries), damage to organs other than the brain may contribute to morbidity (*Scott et al., 2006*). Therefore, we provisionally use the term traumatic injury to refer to the primary injury. One goal of this study is to identify the injured body part or parts that cause death within 24 hr.

We quantify death following traumatic injury by determining the percentage of flies that die within 24 hr of the primary injury, which we define as the mortality index at 24 hr ($MI_{24}$). Previously, we found that genotype and age at the time of traumatic injury affect the $MI_{24}$ (*Katzenberger et al., 2013*). Younger flies have a lower $MI_{24}$ than older flies, suggesting that aging-related processes promote death following traumatic injury. In addition, genotype can affect the $MI_{24}$ many fold, indicating the existence of genes that suppress or enhance the secondary injury mechanisms that cause death following traumatic injury. We also found that the innate immune response is activated shortly after primary injuries. In flies, the Toll and Immune deficiency (Imd) innate immune response pathways are responsible for defense against pathogens such as bacteria (*Lemaitre and Hoffmann, 2007*).

Both pathways upregulate the transcription of antimicrobial peptide (AMP) genes, which encode small, secreted peptides that contribute to the elimination of pathogens. The innate immune response pathways are also activated in response to various types of stress, including oxidative stress and tissue damage. Here, we further investigate the roles of aging, genotype, and the innate immune response in mortality following traumatic injury.

We performed a genome-wide association (GWA) study that implicates specific genes in affecting the probability of death following traumatic injury in young flies. Several of the genes have functions related to septate junctions, which are similar to tight junctions in vertebrates (*Furuse and Tsukita, 2006*). Septate junctions and tight junctions serve as barriers in the intestine, brain, and other tissues that prevent pathogen invasion and restrict the paracellular transport of materials. These junctions are constructed of transmembrane proteins such as Claudins, which interact between neighboring cells, and intracellular proteins such as PDZ (PSD-95, Discs-large, ZO-1) domain proteins, which interact with the cytoplasmic tail of transmembrane proteins. Mutation of the septate junction-associated, PDZ domain-containing protein Big Bang (BBG) permits bacteria from the intestinal lumen to cross the intestinal epithelial barrier and activate the innate immune response (*Bonnay et al., 2013*). In addition, aging-related death in flies is highly correlated with intestinal barrier dysfunction and activation of the innate immune response (*Rera et al., 2012*). In light of these links among tissue barrier dysfunction, the innate immune response, and aging-related death, we investigated the role of tissue barrier dysfunction in death following traumatic injury. Our findings indicate that traumatic injury to the brain is a major cause of death in our model and that mortality from brain injury is dependent on genetic and environmental effects on intestinal barrier permeability.

## Results

### The probability of death following traumatic injury is a quantitative trait

To investigate the role of genotype in determining the $MI_{24}$, we analyzed the *D. melanogaster* Genetic Reference Panel (DGRP), a collection of wild-type, fully sequenced, isogenic fly lines (called RAL lines) (*Mackay et al., 2012*). *Figure 1A* shows the $MI_{24}$ data for 179 RAL lines that were treated with the standard injury protocol. 60 young flies (0–7 day old) were placed in a vial and subjected to four strikes from the HIT device with 5 min between strikes. Following a 10 min recovery period after the last strike, flies were transferred to a new vial containing molasses food and were incubated at 25°C. The number of dead flies was counted after 24 hr. To control for death not due to injuries during this time, a vial of flies not subjected to injury was equivalently processed. Every experiment consisted of at least three independent trials, and the $MI_{24}$ represents the average percent death for flies with injuries minus the average percent death for flies without injuries. We found that the $MI_{24}$ had a continuous distribution among the RAL lines, over a wide range from $6.7 \pm 0.8$ to $57.5 \pm 1.7$ (*Figure 1A* and *Supplementary file 1*). Similarly, we found that the $MI_{24}$ had a continuous distribution among a collection of 53 wild-type African lines, from $24.8 \pm 9.8$ to $68.0 \pm 6.4$ (*Figure 1—figure supplement 1* and *Supplementary file 2*). These data indicate that the $MI_{24}$ is influenced by genotype and that genetic variants affecting this parameter occur among natural populations of *Drosophila*.

To further assess the effect of genotype on the $MI_{24}$, we crossed the RAL line that had the highest $MI_{24}$ (RAL892) to other RAL lines and determined the $MI_{24}$ of 0–7 day old progeny. We found that progeny from crosses between RAL892 and RAL lines with a low $MI_{24}$ (RAL352, RAL907, and RAL774) had an intermediate $MI_{24}$ (*Figure 1C*). In contrast, progeny from crosses between RAL892 and RAL lines with a high $MI_{24}$ (RAL707, RAL73, RAL799, and RAL161) maintained a high $MI_{24}$. Variation in the $MI_{24}$ could be due to many genes or environmental factors. However, the fly lines were cultured under the same conditions (temperature, humidity, diet, light/dark cycle, and density), which limited the contribution of environmental factors. Thus, the continuous distribution over a wide range of the $MI_{24}$ among wild-type fly lines and the intermediate $MI_{24}$ of progeny from crosses between fly lines with significantly different $MI_{24}$s suggest that the probability of death following traumatic injury is a quantitative trait affected by many genes (*Falconer and Mackay, 1996*).

### GWA analysis identifies genes associated with the probability of death following traumatic injury

To identify genes that affect the $MI_{24}$, we carried out GWA analysis using the $MI_{24}$ data shown in *Figure 1A* and ~2.5 million single nucleotide polymorphisms (SNPs) among the RAL lines

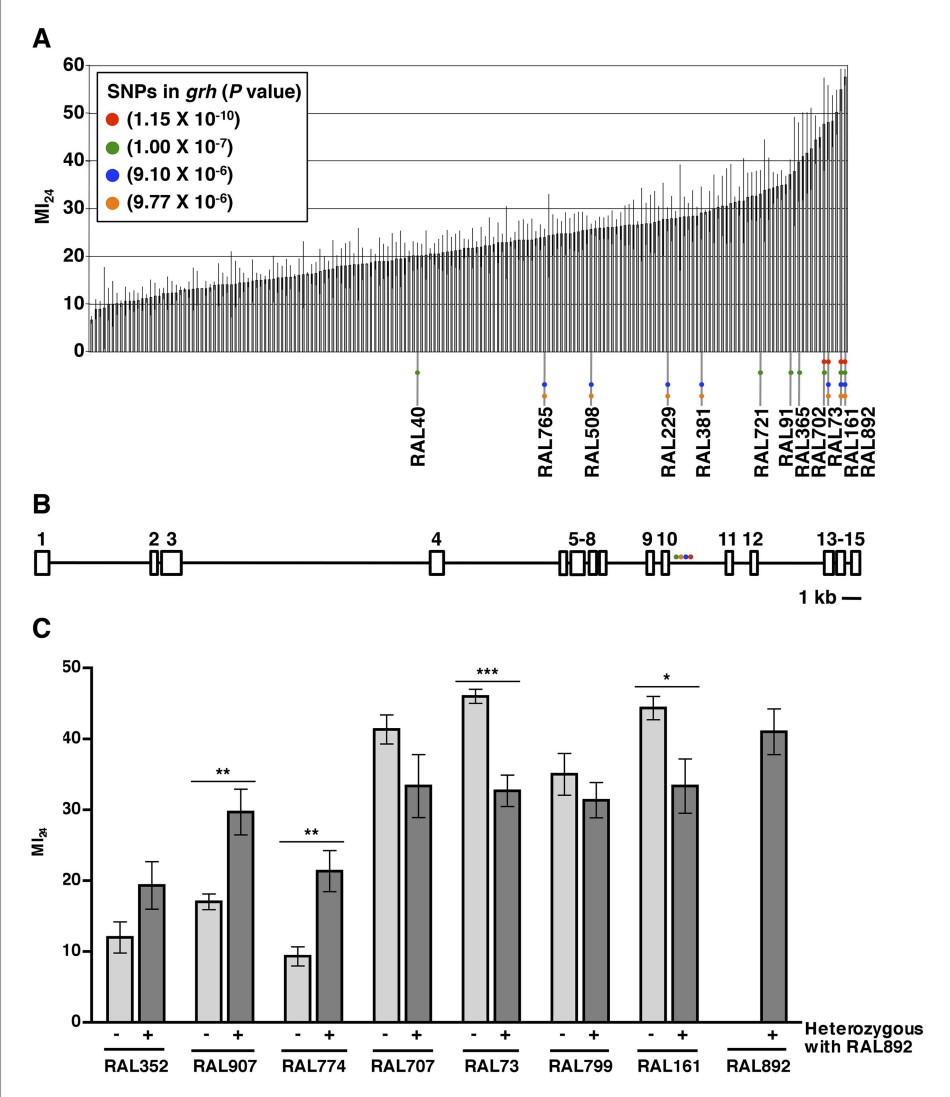

**Figure 1**. The $MI_{24}$ greatly varies among 0–7 day old RAL flies and is associated with SNPs in *grh*. (**A**) Average and standard deviation of the $MI_{24}$ for 179 RAL lines. ***Supplementary file 1*** lists $MI_{24}$ values for each of the RAL lines. Colored dots represent SNPs in *grh* associated with the $MI_{24}$ and are used to indicate the RAL lines that contain the SNPs. (**B**) Schematic diagram of the intron-exon structure of *grh* with the location of the four SNPs associated with the $MI_{24}$ (***St Pierre et al., 2014***). Numbered boxes indicate exons and lines indicate introns. Note that spacing of the colored dots is not drawn to scale. (**C**) SNPs affect the $MI_{24}$. Average and standard error of the mean (SEM) of the $MI_{24}$ for RAL lines (light gray bars) and progeny from crosses between RAL lines and RAL892 (dark gray bars). *p < 0.05, **p < 0.01, and ***p < 0.001, one-tailed *t* test comparison.

The following figure supplement is available for figure 1:

**Figure supplement 1**. The $MI_{24}$ greatly varies among 0–7 day old wild-type African lines.

(***Mackay et al., 2012***). This analysis revealed that 216 unique SNPs located in or near 98 genes were associated with the $MI_{24}$ at a discovery significance threshold of p < $10^{-5}$ (***Supplementary file 3***). However, despite the small p-values, some of the associations may be false positives because the minor allele frequency cut-off of the DGRP Freeze 1 algorithm was 4 lines, allowing the p-value to be driven by a few extreme lines. Reanalysis using the DGRP Freeze 2 algorithm that has a minor allele frequency cut-off of 10 lines revealed significant associations of SNPs in only 10 of the 98 genes (***Huang et al., 2014***). The discrepancy between the Freeze 1 and 2 analyses is illustrated by SNPs in

*grainyhead* (*grh*), which were significantly associated with the $MI_{24}$ in the Freeze 1 analysis but were not identified in the Freeze 2 analysis because they occurred in fewer than 10 lines (*Figure 1A* and *Supplementary file 1*). At the time that we obtained the $MI_{24}$ data for the RAL lines, the Freeze 2 algorithm had not been developed, so we moved forward based on the Freeze 1 data, initially focusing on *grh* because it contained the SNP that was most significantly associated with the $MI_{24}$ (p = $1.15 \times 10^{-10}$) as well as three other significant SNPs. The four SNPs in *grh* were located in a 523 bp region of intron 10 suggesting that they have similar effects on the regulation of *grh* expression (*Figure 1B*) (*St Pierre et al., 2014*). Alternatively, since three of the four SNPs (red, blue, and yellow dots in *Figure 1A*) are shared by three lines (RAL73, RAL161, and RAL892), linkage disequilibrium may account for their significant association with the $MI_{24}$. These data indicate that flies carrying particular *grh* alleles are more likely to die within 24 hr of a traumatic injury than flies lacking these alleles.

*Grh* encodes a transcription factor crucial for many aspects of development, including epithelial barrier formation (*Nüsslein-Volhard et al., 1984*; *Paré et al., 2012*). In humans, one of the three *grh* orthologs, *Grainyhead-like 2* (*Grhl2*) activates the expression of *claudin* genes, and in mice, *Grhl3* knockout reduces the expression of *claudin* genes (*Yu et al., 2008*; *Werth et al., 2010*; *Senga et al., 2012*). In flies, misexpression of *grh* in a tissue that normally lacks septate junctions is sufficient to induce expression of septate junction proteins (*Narasimha et al., 2008*). Thus, we hypothesized that the four SNPs in *grh* affect the function of septate junctions by altering the expression of genes encoding septate junction proteins. In support of this hypothesis, SNPs in *bbg* and *scribbled* (*scrib*), which encode PDZ domain-containing, septate junction-associated proteins, were also associated with the $MI_{24}$ (*Supplementary files 3, 4*) (*Bilder and Perrimon, 2000*; *Bonnay et al., 2013*). *bbg* remained significantly associated with the probability of death (p = $2.36 \times 10^{-6}$) when the data from *Figure 1A* were reanalyzed using the DGRP Freeze 2 algorithm (*Huang et al., 2014*). Additional support for the hypothesis comes from the finding that direct mechanical damage to the brain in rodent TBI models causes disruption of the intestinal barrier and a decrease in expression of tight junction proteins (*Hang et al., 2003*; *Feighery et al., 2008*; *Jin et al., 2008*; *Bansal et al., 2009, 2010*). Lastly, gastrointestinal dysfunction is a common complication in TBI patients, and disruption of intestinal tight junction barriers can trigger systemic diseases (*Krakau et al., 2006*; *Suzuki, 2013*). Thus, we tested this hypothesis by examining the permeability of tissue barriers following traumatic injury.

## Traumatic injury causes intestinal barrier dysfunction

Functionality of intestinal barrier can be ascertained in flies using a dye permeability assay in which flies are fed a nonabsorbable blue dye (*Rera et al., 2011, 2012*). If the intestinal barrier is functional, the dye remains in the digestive tract (*Figure 2A*). In contrast, if the intestinal barrier is disrupted, the dye crosses the barrier into the hemolymph and disperses throughout the body, a process referred to as 'Smurfing'. Hemolymph is extracellular fluid in the open circulatory system of insects that contacts all internal organs and carries substances such as nutrients and metabolic waste to and away from cells, respectively (*Handke et al., 2013*). We found that treatment of 0–7 day old $w^{1118}$ flies (a common laboratory strain) with the standard injury protocol caused $23.3 \pm 2.1\%$ of the flies to Smurf within 24 hr of the primary injury (*Figure 2C*), whereas only $0.5 \pm 0.2\%$ of untreated flies Smurfed in the same time period. These data indicate that traumatic injury increases the permeability of the intestinal barrier in flies.

To determine whether direct injury to the brain affects intestinal permeability, we inflicted closed-head injuries and monitored Smurfing as a reporter of intestinal permeability. We found that brain injury caused by compressing the head of 0–7 day old $w^{1118}$ flies from eye-to-eye using forceps was sufficient to cause Smurfing within 24 hr (*Figure 2B*). Of the 540 treated flies, 15.4% Smurfed and died within 24 hr of the primary injury, whereas only 0.5% Smurfed but did not die and 4.6% died but did not Smurf. In contrast, of the 540 untreated flies, 0.7% Smurfed and died within 24 hr, none Smurfed but did not die, and 0.4% died but did not Smurf. These data are consistent with the observed link between brain injury and intestinal barrier dysfunction in rodents, and they support the conclusion that increased intestinal permeability of flies subjected to the HIT device is due to brain injury (*Hang et al., 2003*; *Feighery et al., 2008*; *Jin et al., 2008*; *Bansal et al., 2009, 2010*). In addition, these data suggest a causal link between increased intestinal permeability and death following TBI.

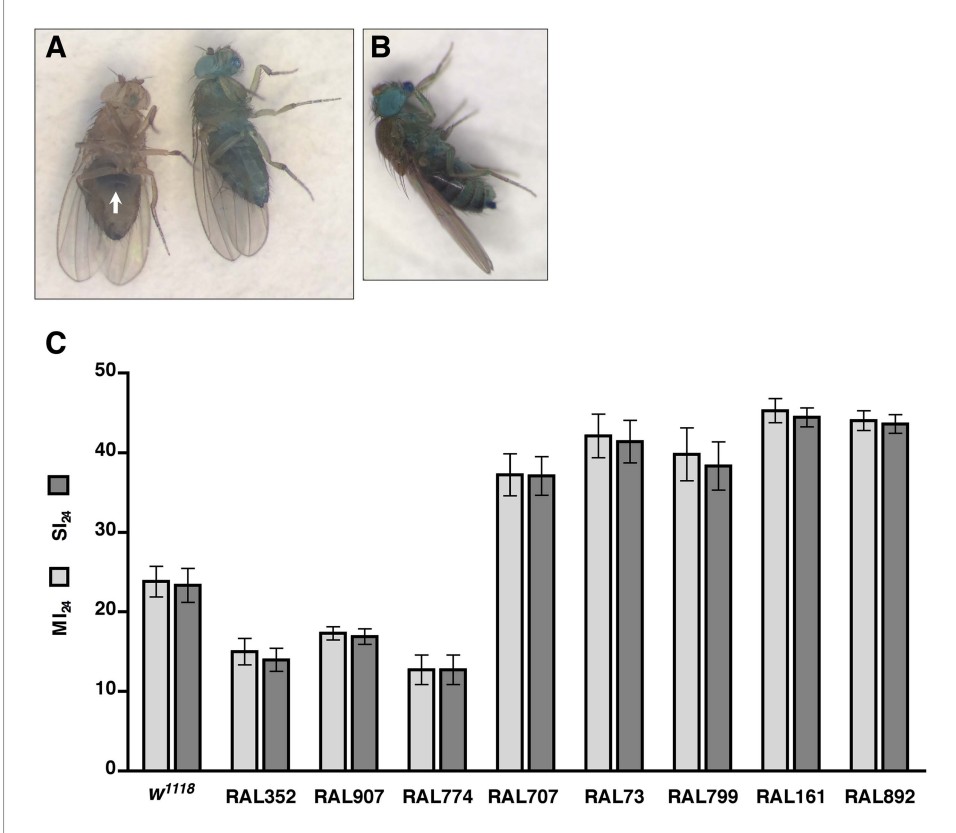

**Figure 2**. Traumatic injury causes intestinal barrier dysfunction. (**A**) Flies that were fed molasses food with blue dye. In flies without traumatic injury (left), the dye was confined to the gut (arrow). In some flies with traumatic injury (right), the dye leaked out of the intestine into the hemolymph and dispersed throughout the body, producing a 'Smurf' phenotype (*Rera et al., 2011*; *Rera et al., 2012*). (**B**) A fly that was fed molasses food with blue dye and received brain injury from head compression. (**C**) Average and SEM of the $MI_{24}$ (light gray bars) and $SI_{24}$ (dark gray bars) for the indicated fly lines. The $MI_{24}$ and $SI_{24}$ were not significantly different for any of the fly lines (p > 0.32, one-tailed *t* test). The correlation coefficient (r) between the $MI_{24}$ and $SI_{24}$ was 1.0.

The following figure supplement is available for figure 2:

**Figure supplement 1**. Incapacitated flies had a significantly higher $MI_{24}$ than non-incapacitated flies (p = 0.007, one-tailed *t* test).

## Traumatic injury causes blood-eye barrier (BEB)/blood–brain barrier (BBB) dysfunction

In mammals, TBI not only disrupts the intestinal barrier but also the BBB (*Alves, 2014*). Therefore, we used a fluorescence assay to examine the effect of traumatic injury on integrity of the BEB as a reporter of the BBB (*DeSalvo et al., 2011*; *Pinsonneault et al., 2011*). Septate junctions are essential for creating the BEB, which restricts the transport of molecules between the retina and the hemolymph (*Banerjee et al., 2008*). We used intra-thoracic injection to introduce tetramethylrhodamine-conjugated dextran molecules (MW = 10,000) into the hemolymph of the fly. If the BEB is intact, the molecules accumulate along the border of the eye forming a hemolymph exclusion line (*Figure 3A*) (*Pinsonneault et al., 2011*). In contrast, if the BEB is disrupted, the molecules cross the barrier and disperse throughout eye (*Figure 3B*). We subjected 1–4 day old $w^{1118}$ flies to the standard injury protocol, injected them in the thorax with fluorescent molecules, waited 2 hr, and examined the pattern of fluorescence in the eyes. We found that relative to untreated flies, a significantly greater percentage of HIT device-treated flies had fluorescence throughout the eye, indicating that the BEB is disrupted following traumatic injury (*Figure 3C*). Furthermore, because

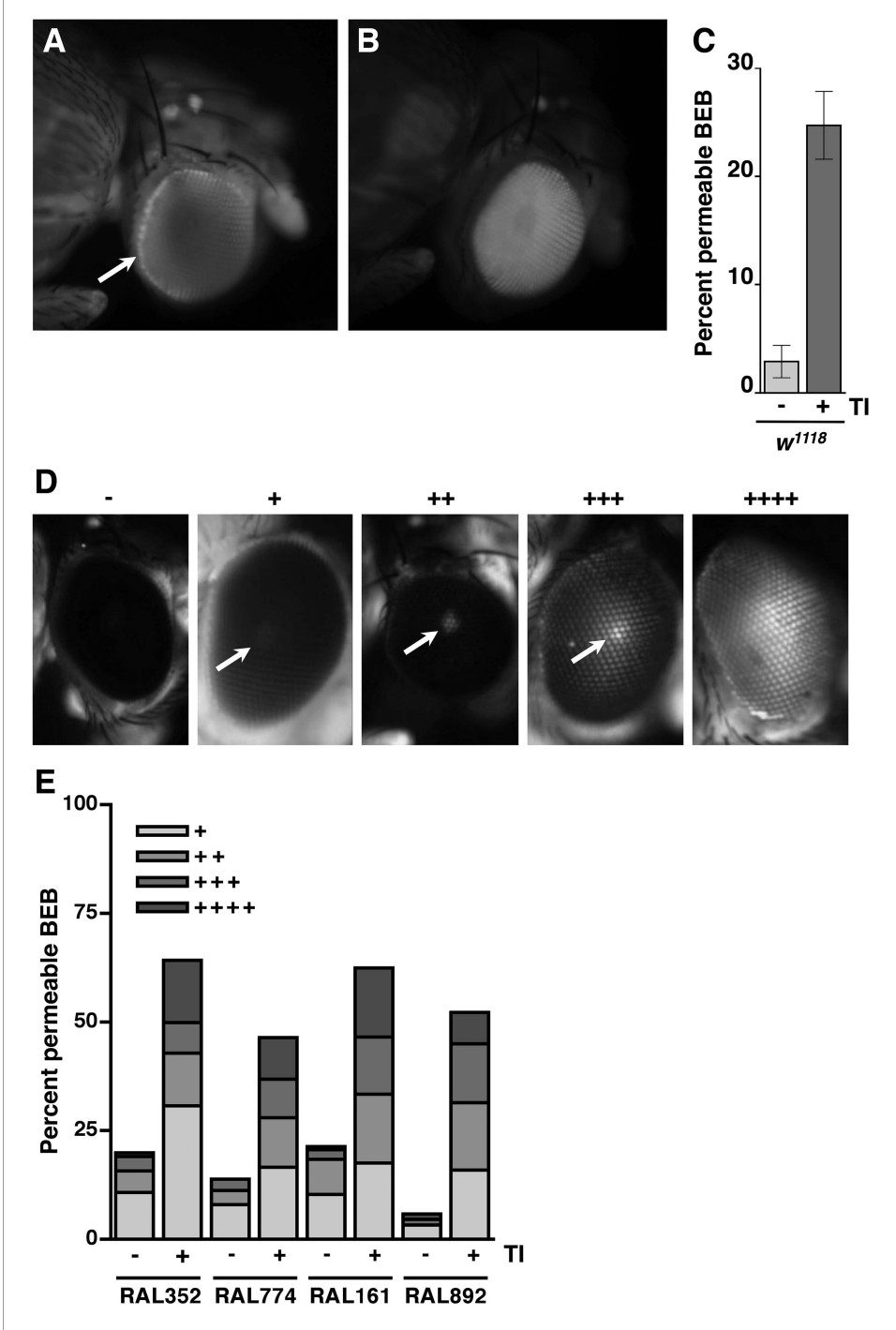

**Figure 3**. Traumatic injury causes BEB disruption. (**A**) A $w^{1118}$ fly without traumatic injury that was injected with tetramethylrhodamine-dextran molecules. Note the accumulation of fluorescence at the border of the eye (arrow), which reflects an intact BEB. (**B**) A $w^{1118}$ fly with traumatic injury that was injected with tetramethylrhodamine-dextran molecules. Note the fluorescence throughout the eye, which reflects BEB permeability. (**C**) Average and SEM of the percent of HIT device-treated (+) and untreated (−) $w^{1118}$ flies with a permeable BEB. Traumatic injury (TI) significantly increased the percent of flies with a permeable BEB (p = 0.0017, one-tailed $t$ test). (**D**) The dye penetration scale from − to ++++ for RAL flies. Arrows indicate the fluorescent pseudopupil. (**E**) The percent of flies with (+) or without (−) traumatic injury in each scale category for the indicated RAL lines. At least 85 flies were examined for each condition.

permeability of the BEB is a reporter of permeability of the BBB, these data suggest that traumatic injury also causes BBB disruption in flies (*DeSalvo et al., 2011*).

## Intestinal barrier dysfunction correlates with the probability of death following traumatic injury

To investigate a causal link between intestinal barrier dysfunction and death following traumatic injury, we determined the overlap between flies that Smurfed and flies that died within 24 hr of the primary injury. In the case of $w^{1118}$ flies, the percentage of flies that Smurfed within 24 hr of the primary injury, which we define as the Smurfing Index at 24 hr ($SI_{24}$), was statistically similar to the $MI_{24}$ (*Figure 2C*). Moreover, there was almost complete overlap between flies that Smurfed and flies that died; less than 1% of flies Smurfed but did not die within 24 hr and less than 1% of flies that did not Smurf died within 24 hr.

To determine the generality of these findings, we examined eight RAL lines: three of which had a low $MI_{24}$ (RAL352, RAL907, and RAL774) and five of which had a high $MI_{24}$ (RAL707, RAL73, RAL799, RAL161, and RAL892). In all cases, we found that there was no significant difference between the $SI_{24}$ and $MI_{24}$ for each fly line (*Figure 2C*). The almost perfect correlation between the $SI_{24}$ and $MI_{24}$ suggests that intestinal barrier dysfunction is closely linked with death following traumatic injury.

As we have previously reported, about 10% of flies subject to a single strike from the HIT device are temporarily incapacitated, lying motionless on their back or side with no evident physical damage before gradually recovering motor activity within 5 min (*Katzenberger et al., 2013*). This phenotype is similar to the symptoms of a concussion in humans (*Giacino et al., 2014*). It is also similar to temporary paralysis observed in bang-sensitive *Drosophila* mutants following mild mechanical shock that disrupts normal electrical activity in the brain (*Fergestad et al., 2008*; *Parker et al., 2011*; *Burg and Wu, 2012*). These observations are consistent with the idea that at least a fraction of the flies subjected to the HIT device suffer a brain injury that temporarily disturbs normal neuronal function resulting in temporary paralysis. These observations also raise the question of whether this presumptive brain injury contributes to the observed mortality of treated flies. To address this question, we determined the correlation between temporary incapacitation and the $MI_{24}$ following traumatic injury. Individual flies were subjected to a single strike from the HIT device and were scored as incapacitated if they were motionless immediately after injury. As previously observed, incapacitated flies did not die immediately (*Katzenberger et al., 2013*). Of the 600 flies examined, 64 (10.7%) were incapacitated, and all, except one, recovered mobility within 5 min. Nonetheless, incapacitated flies had an ~fivefold higher $MI_{24}$ than non-incapacitated flies, indicating that injuries that cause incapacitation also contribute significantly to the cause of death within 24 hr (*Figure 2—figure supplement 1*). Although we cannot rule out other possibilities, these data taken together with data in *Figure 2* are consistent with a model in which death following traumatic injury inflicted by the HIT device is dependent on intestinal barrier dysfunction, which is evoked by damage to the brain via an unknown mechanism.

## BEB/BBB dysfunction does not correlate with the probability of death following traumatic injury

To investigate a causal link between BEB/BBB and death following traumatic injury, we used the RAL lines to examine the correlation between BEB dysfunction and the $MI_{24}$. Because the RAL lines have red eyes, the BEB disruption phenotype is different from that described earlier for $w^{1118}$ flies, necessitating a modification of the BEB protocol (*DeSalvo et al., 2011*). Flies with an intact BEB had eyes with no fluorescence, whereas flies with a permeable BEB had eyes with a fluorescent pseudopupil that ranged in intensity. Using the scale shown in *Figure 3D*, we qualitatively scored the intensity of fluorescence in various RAL lines before and after subjecting flies to injury. We found that the percent of flies with a leaky BEB following traumatic injury, that is, those scored + to ++++, was comparable among different RAL lines, as was the distribution of flies among the scale categories, irrespective of whether a particular line had a low $MI_{24}$ (RAL352 and RAL774) or a high $MI_{24}$ (RAL161 and RAL892) (*Figure 3E*). In addition, the percent of RAL352 and RAL774 flies with BEB dysfunction following injury was substantially higher than their respective $MI_{24}$ values. These data indicate that BEB/BBB dysfunction does not correlate with the probability of death following traumatic injury.

## Bacteria and glucose levels increase in the hemolymph shortly after traumatic injury

The correlation between intestinal barrier disruption and the $MI_{24}$ suggested the hypothesis that death following traumatic injury is triggered by paracellular leakage of factors such as bacteria or food components from the intestinal lumen to the hemolymph. The *Drosophila* gut commonly contains bacterial species in the *Lactobacillus* and *Acetobacter* genera (*Buchon et al., 2013*). To determine if traumatic injury permits bacteria to leak across the impaired intestinal barrier, we quantified bacterial levels in the hemolymph. We extracted hemolymph from 0–7 day old $w^{1118}$ flies 1 hr after they were subjected to the standard injury protocol and determined the bacterial count by spreading a given amount of hemolymph on LB plates (*Liu et al., 2012*). We found that HIT device-treated flies had >400-fold more bacteria in the hemolymph than untreated flies (*Figure 4A* and *Figure 4—figure supplement 1*).

To determine if traumatic injury permits glucose to leak across the impaired intestinal barrier, we determined the concentration of glucose in the hemolymph. Hemolymph was extracted from 0–7 day old $w^{1118}$ flies at various time points after treatment with the standard injury protocol and the glucose concentration was determined using a colorimetric-based enzymatic assay (*Tennessen et al., 2014*). We found that the hemolymph glucose concentration of injured flies was significantly higher than that of untreated flies between 2 and 8 hr after injury (*Figure 4B*). Collectively, the bacteria and blood glucose data indicate that traumatic injury disrupts paracellular barriers formed by septate junctions, allowing the escape of factors at least as large as a bacterium from the intestinal lumen into the hemolymph.

To address whether ingested food is the source of increased glucose in the hemolymph after traumatic injury, we determined the glucose concentration of hemolymph from flies fed molasses food or water after treatment with the standard injury protocol. We found that traumatic injury significantly increased the glucose concentration of hemolymph of flies fed molasses food but not flies fed water (*Figure 4C*) indicating that ingested molasses food is the source of increased glucose in the hemolymph following traumatic injury.

## Bacteria do not affect the probability of death following traumatic injury

As leakage of bacteria from the intestine could contribute to death following traumatic injury, we tested this possibility by eliminating endogenous bacteria in the gut and elsewhere by feeding flies a mixture of antibiotics in molasses food. It was previously shown that the mixture of antibiotics does not interfere with Imd pathway activation and is not toxic to flies (*Liu et al., 2012*). After feeding antibiotics to 0–2 day old flies for 5 day, we treated the resulting 5–7 day old flies with the standard injury protocol. Some of the flies were used to determine the $MI_{24}$, and others were used to determine the effectiveness of the antibiotic treatment. For $w^{1118}$ and RAL fly lines, PCR analysis of bacterial 16S rDNA levels using primers that recognize most bacterial species revealed that antibiotic treatment eliminated the endogenous bacteria (*Figure 5B*) (*Weisburg et al., 1991*; *Liu et al., 2012*; *Wong et al., 2013*). Colony counts of whole fly extracts spread on LB plates yielded the same conclusion (*Figure 5—figure supplement 1*). Nonetheless, we found that antibiotics did not significantly affect the $MI_{24}$ (*Figure 5A*). Thus, bacteria do not significantly contribute to mortality following traumatic injury. Similarly, we found that the $SI_{24}$ of antibiotic-fed flies was not significantly different from the $SI_{24}$ of flies without antibiotics (*Figure 5C*) indicating that bacteria are also not involved in primary or secondary mechanisms that cause intestinal barrier dysfunction.

## Activation of the innate immune response predicts but does not contribute to death following traumatic injury

We previously observed that expression of the innate immune response is activated in flies following traumatic injury (*Katzenberger et al., 2013*). Leakage of bacteria across the intestinal barrier would be one way that traumatic injury could trigger the innate immune response. If so, this response should be dampened in antibiotic-fed flies. We used qRT-PCR to quantify AMP gene expression following traumatic injury in flies treated with antibiotics compared to controls. We found that 2 hr after treatment of 0–7 day old $w^{1118}$ flies with the standard injury protocol, both antibiotic-fed flies and flies without antibiotics had higher levels of expression of the AMP genes *Attacin C* (*AttC*), *DiptB* (*Diptericin B*), and *Metchnikowin* (*Mtk*) than equivalently treated flies that were not subjected to the

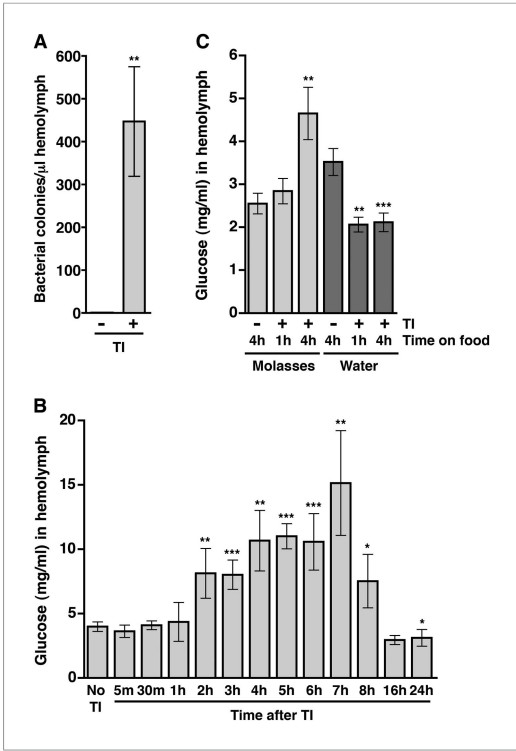

**Figure 4**. Traumatic injury causes an increase in the amount of bacteria and glucose in the hemolymph of $w^{1118}$ flies. (**A**) Average and SEM of the number of bacterial colonies per microliter of hemolymph from flies without (−) or with (+) traumatic injury (TI). Flies without traumatic injury had 0.8 ± 0.9 bacterial colonies per microliter of hemolymph. (**B**) Average and SEM of glucose concentration at times after traumatic injury. A significant increase in glucose concentration occurred between 2 and 8 hr, and a significant decrease in glucose concentration occurred at 24 hr. (**C**) Average and SEM of glucose concentration in flies fed either molasses food or water for the indicated amount of time after traumatic injury. Molasses food significantly increased the glucose concentration. In contrast, water significantly decreased the glucose concentration. *p < 0.05, **p < 0.01, and ***p < 0.001, one-tailed *t* test comparison between flies without (−) and with (+) traumatic injury.

The following figure supplement is available for figure 4:

**Figure supplement 1**. Traumatic injury causes bacteria to leak into the hemolymph.

standard injury protocol (**Figure 6A** and **Figure 6—figure supplements 1, 3**). Similar results were observed for some of the RAL lines. These data indicate that activation of the innate immune response following traumatic injury is not solely due to a bacteria-dependent mechanism but is triggered by other injury-associated factors as well.

We also found that the antibiotic-fed flies had significantly lower levels of expression of *AttC*, *DiptB*, and *Mtk* than flies without antibiotics (**Figure 6B** and **Figure 6—figure supplements 2, 4**). Thus, antibiotics reduce the level of activation of the innate immune response but do not affect the $MI_{24}$, indicating that death following traumatic injury is not influenced by activation of the innate immune response. On the other hand, the level of AMP gene expression does correlate with the $MI_{24}$. For example, the correlation coefficient (r) between *AttC* expression and the $MI_{24}$ for the eight RAL lines examined was 0.75 (p = 0.03) for antibiotic-fed flies and 0.74 (p = 0.03) for flies without antibiotics. The respective r-values for *DiptB* were 0.52 (p = 0.19) and 0.51 (p = 0.20) and for *Mtk* were 0.81 (p = 0.02) and 0.63 (p = 0.09). These data indicate that the level of expression of some AMP genes, for example, *AttC* and possibly *Mtk*, is predictive of death following traumatic injury. Possibly, AMP gene expression level may somehow reflect the extent of intestinal barrier permeability following injury and thus be a predictor of subsequent mortality.

## Molasses food ingestion after traumatic injury causes intestinal barrier dysfunction and death

As mortality following traumatic injury did not appear to be influenced by leakage of bacteria through a disrupted intestinal barrier, we went on to inquire whether leakage of an ingested food component influenced mortality by reducing the amount of food in the gut before or after traumatic injury. For the 'before' treatment, we cultured 0–6 day old $w^{1118}$ flies in vials with water-soaked filter paper for 24 hr, subjected flies to the standard injury protocol, transferred them to vials containing molasses food, and determined the $MI_{24}$. We found that the $MI_{24}$ of flies fed water before traumatic injury did not differ significantly from that of molasses food-fed flies (**Figure 7A**). For the 'after' treatment, flies were cultured on molasses food prior to the standard injury protocol and then transferred to vials containing water-soaked filter paper. We found that feeding the flies water rather than molasses food after traumatic injury significantly reduced the $MI_{24}$. Feeding water rather than molasses food after injury also reduced the $MI_{24}$ of RAL lines with a low (RAL774) or high (RAL707) $MI_{24}$ (**Figure 7B**). These data indicate that death following traumatic injury is

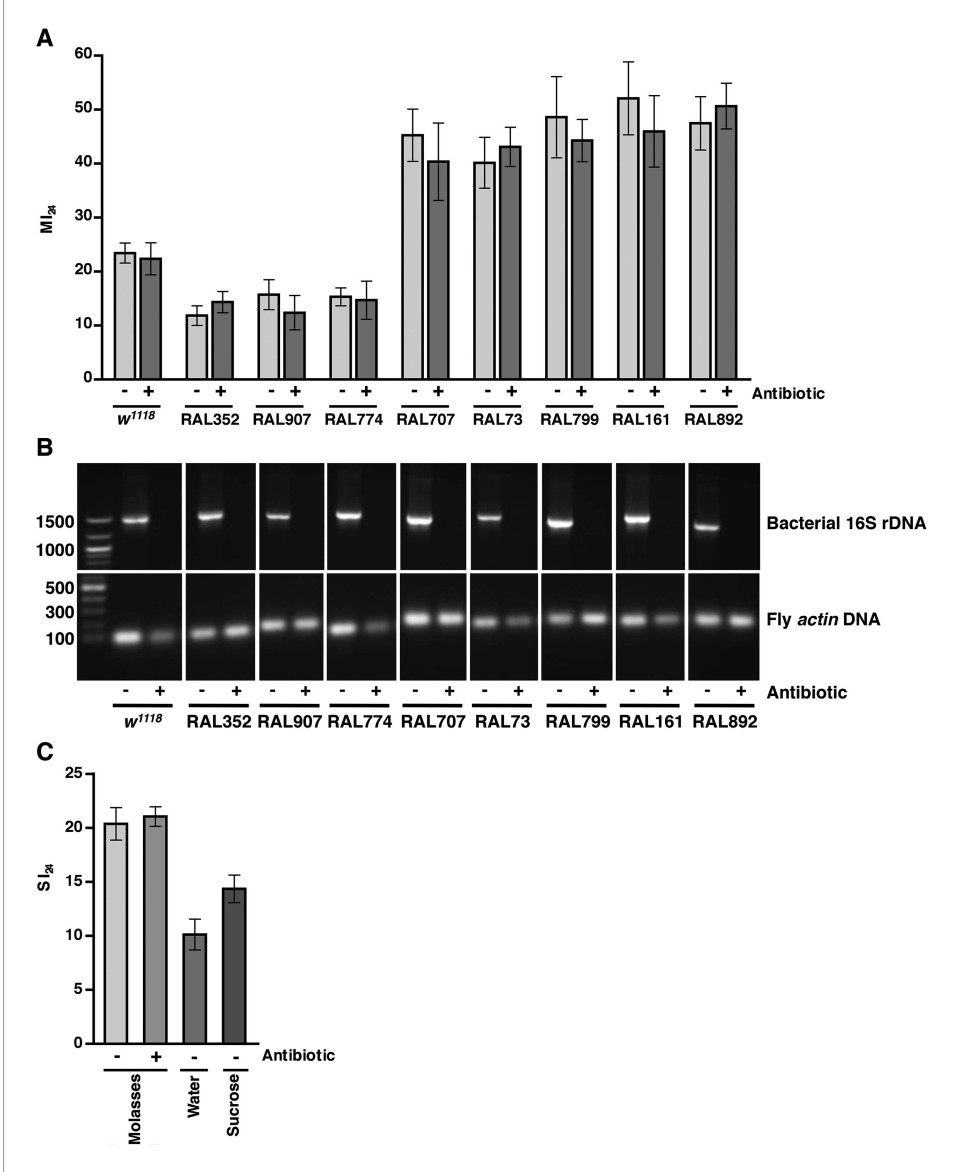

**Figure 5**. Endogenous bacteria do not affect the probability of death following traumatic injury. (**A**) Average and SEM of the $MI_{24}$ for the indicated fly lines fed for 5 day on molasses food (light gray bars) or on molasses food containing antibiotics (dark gray bars) before being subjected to the standard injury protocol. (**B**) Levels of bacteria in fly lines shown in panel **A**, as detected by PCR analysis for bacterial 16S rDNA and fly *actin* as a loading control. DNA extracted from flies fed (+) or not fed (−) antibiotics was used as a template. Indicated on the left are DNA size markers in basepairs. (**C**) Average and SEM of the $SI_{24}$ for $w^{1118}$ flies with the indicated treatments. Flies fed antibiotics had an $SI_{24}$ that was not significantly different than flies without antibiotics (p = 0.36, one-tailed *t* test). Flies fed water after the primary injury had an $SI_{24}$ that was significantly lower than flies fed molasses food after the primary injury (p < 0.0001, one-tailed *t* test). Flies fed 1.2 M sucrose after the primary injury had an $SI_{24}$ that was significantly higher than flies fed water after the primary injury (p = 0.0027, one-tailed *t* test).

The following figure supplement is available for figure 5:

**Figure supplement 1**. Antibiotic treatment of flies eliminates endogenous bacteria.

dependent on a secondary injury mechanism that involves ingestion of molasses food after the primary injury. However, the present data do not rule out the possibility that water provides protection against death following traumatic injury. In either case, since death following traumatic

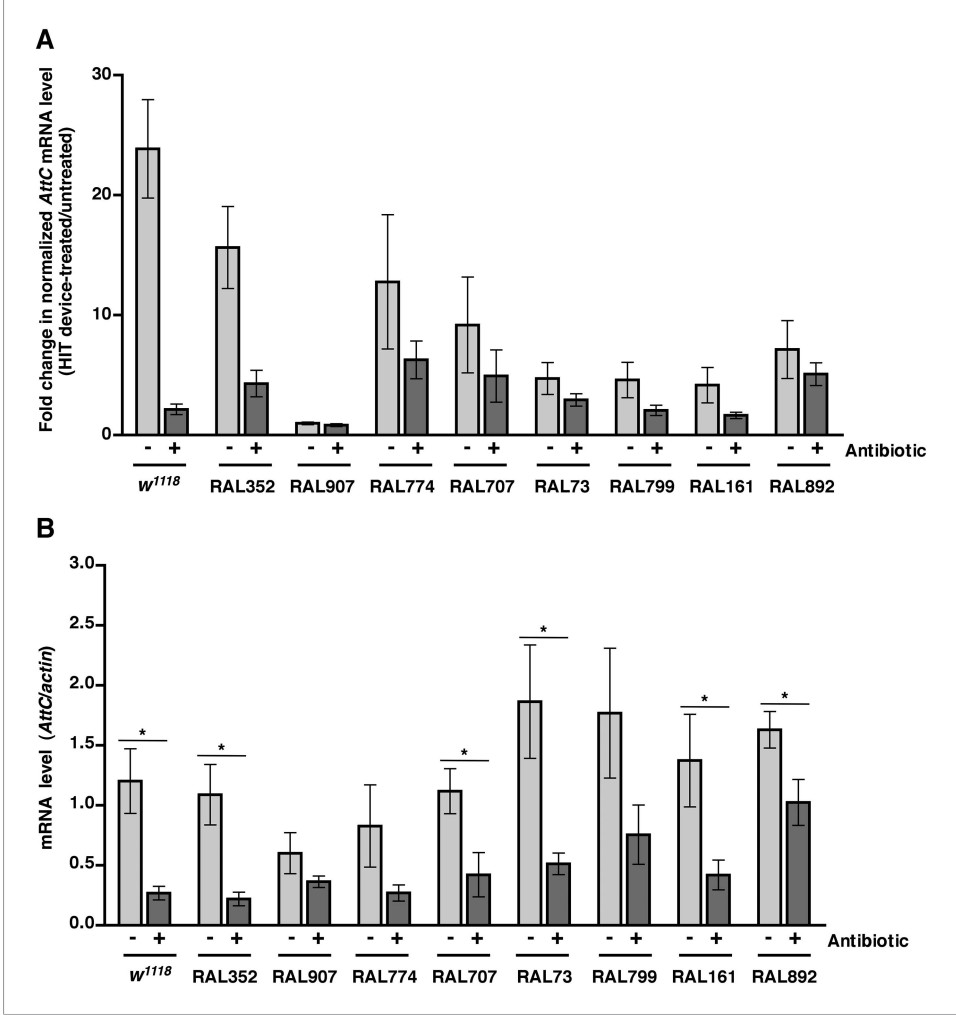

**Figure 6**. Analyses of the role that bacteria play in activation of the innate immune response by traumatic injury and the role that the level of activation of the innate immune response plays in causing death following traumatic injury. (**A**) Average and SEM of *AttC* expression normalized to *actin* expression in HIT device-treated flies relative to untreated flies. This analysis was performed with flies fed food without antibiotics (light gray bars) or with antibiotics (dark gray bars). Expression levels were determined 2 hr after treatment with the standard injury protocol. Analogous data are shown for *DiptB* and *Mtk* in **Figure 6—figure supplements 1, 3**, respectively. (**B**) Average and SEM of *AttC* expression normalized to *actin* expression in antibiotic-fed flies (dark gray bars) and flies without antibiotics (light gray bars) 2 hr after treatment with the standard injury protocol. Analogous data are shown for *DiptB* and *Mtk* in **Figure 6—figure supplements 2, 4**, respectively. *p < 0.05, one-tailed t test.

The following figure supplements are available for figure 6:

**Figure supplement 1**. Level of *DiptB* expression normalized to *actin* expression in HIT device-treated flies relative to untreated flies for flies without antibiotics (light gray bars) and antibiotic-fed flies (dark gray bars).

**Figure supplement 2**. Level of *DiptB* expression normalized to *actin* expression 2 hr after treatment with the standard injury protocol in flies without antibiotics (light gray bars) antibiotic-fed flies (dark gray bars).

**Figure supplement 3**. Level of *Mtk* expression normalized to *actin* expression in HIT device-treated flies relative to untreated flies for flies without antibiotics (light gray bars) and antibiotic-fed flies (dark gray bars).

**Figure supplement 4**. Level of *Mtk* expression normalized to *actin* expression 2 hr after treatment with the standard injury protocol in flies without antibiotics (light gray bars) and antibiotic-fed flies (dark gray bars).

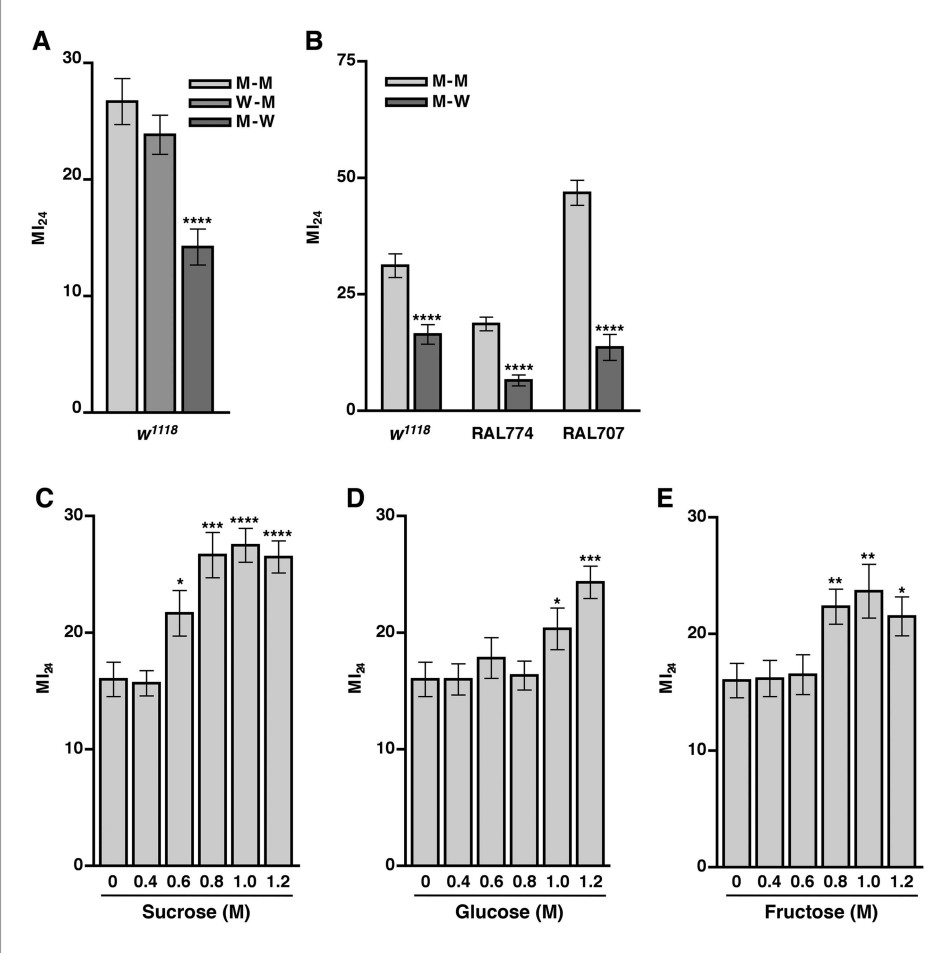

**Figure 7**. Food ingested after the primary injury affects the $MI_{24}$. (**A**) Average and SEM of the $MI_{24}$ for flies fed water (W) or molasses food (M) for 24 hr before or after the primary injury, for example, W–M means water for 24 hr before the primary injury and molasses food for 24 hr after the primary injury. (**B**) Average and SEM of the $MI_{24}$ for flies of the indicated genotype and food treatments. (**C–D**) Average and SEM of the $MI_{24}$ for flies fed the indicated molar (M) concentrations of (**C**) sucrose, (**D**) glucose, and (**E**) fructose for 24 hr after the primary injury. *$p < 0.05$, **$p < 0.01$, ***$p < 0.001$, ****$p < 0.0001$, one-tailed $t$ test.

injury was not completely eliminated by substituting water for molasses food, these data indicate that other independent mechanisms also contribute to mortality after traumatic injury. Consistent with these observations and the correlation between the $SI_{24}$ and mortality after injury, we found that flies fed water after traumatic injury had a significantly lower $SI_{24}$ than molasses food-fed flies (*Figure 5C*). These data indicate that ingestion of molasses food after a primary injury promotes intestinal barrier disruption.

## Sugar ingestion after a primary injury promotes intestinal barrier disruption and death

Our finding that traumatic injury causes glucose levels to increase in the hemolymph suggested that molasses, which is predominantly sucrose but also contains significant amounts of glucose and fructose, is the primary component of molasses food that promotes death following traumatic injury (*Dionex, 2003*). In support of this hypothesis, genes in the hexosamine biosynthesis pathway (HBP), which functions as a sensor and regulator of glucose levels, were associated with the $MI_{24}$ (*Supplementary files 3, 4*) (*Marshall, 2006*). One gene, s*uper sex combs (sxc)*, encodes an O-linked *N*-acetylglucosamine (O-GlcNAc) transferase (OGT) that uses one of the major HBP end products,

UDP-*N*-acetylglucosamine (UDP-GlcNAc), as a substrate for post-translational modification of proteins involved in insulin production and utilization (*Copeland et al., 2008*; *Sinclair et al., 2009*). Another gene, *polypeptide GalNAc transferase 2* (*pgant2*), encodes a polypeptide *N*-acetylgalacto-saminyltransferase that uses the other major HBP end product, UDP-*N*-acetylgalactosamine (UDP-GalNAc), as a substrate for post-translational modification of proteins that remain to be identified (*Zhang and Ten Hagen, 2010*). In addition, SNPs in *microRNA-14* (*mir-14*), which regulates insulin production in neurosecretory cells, and *CG7882*, which encodes a protein similar in sequence to human glucose transporters (GLUTs) that facilitate transport of glucose across plasma membranes, were associated with the $MI_{24}$ (*Thorens and Mueckler, 2010*; *Varghese et al., 2010*).

Consequently, we investigated the possibility that sugar levels contribute to mortality following traumatic injury by determining the $MI_{24}$ of 0–7 day old $w^{1118}$ flies that were transferred to vials with sugar-soaked filter paper (sucrose, glucose, or fructose) following the standard injury protocol. The concentrations of sugars tested were based on molasses food, which is 10% molasses vol/vol (approximately 0.33 M sugar, that is, 0.13 M sucrose, 0.1 M glucose, and 0.1 M fructose) (*Dionex, 2003*). We found that flies cultured on any of the sugars at 0.4 M had an $MI_{24}$ that was not significantly different from the $MI_{24}$ of flies cultured on water (*Figure 7C–E*). In contrast, flies cultured on 0.6 M sucrose, 1.0 M glucose, or 0.8 M fructose had an $MI_{24}$ that was significantly higher than the $MI_{24}$ of flies cultured on water. The effective concentrations of the individual sugars are higher than they are in molasses suggesting that sugars in combination have a synergistic effect on the $MI_{24}$ or that another component of molasses food is important. *Figures 7C–E* also indicate that the effect of sugars on the $MI_{24}$ is saturable. For example, flies cultured on 0.8, 1.0, or 1.2 M sucrose had statistically similar $MI_{24}$ values. We also found that flies fed 1.2 M sucrose after a primary injury had a significantly higher $SI_{24}$ than flies fed water after a primary injury (*Figure 5C*), indicating that sugar ingested after a traumatic injury promotes intestinal barrier disruption. Together, these data indicate that ingestion of sugar beyond a certain threshold after traumatic injury increases the probability of death by exacerbating intestinal barrier dysfunction and that the secondary injury mechanism by which sugar promotes death is saturable.

## Impaired insulin signaling does not correlate with the probability of death following traumatic injury

The increased concentration of glucose in the hemolymph that results from traumatic injury could be exacerbated by reduced transport of glucose from the hemolymph into cells. Therefore, we investigated the effect of traumatic injury on insulin signaling, a major regulator of glucose transport into cells (*Hazelton and Fridell, 2010*). To do this, we examined expression of gene targets of the insulin signaling pathway. Insulin signaling through the insulin receptor (InR) leads to phosphorylation and activation of the Akt kinase and inactivation of the FOXO transcription factor (*Teleman, 2009*). If insulin signaling is impaired, FOXO is activated and expression of FOXO target genes such as *InR*, *Lipase 4* (*Lip4*), *Ecdysone-inducible gene L2* (*Impl2*), and *4E-BP* (*Thor*), increases (*Wang et al., 2005*).

We used qRT-PCR to determine expression levels of FOXO target genes in Smurfed and non-Smurfed 0–7 day old $w^{1118}$ flies 2 hr after treatment with the standard injury protocol and culturing on molasses food. We found that Smurfed flies had a small but significant increase in expression of three of the four FOXO target genes compared with non-Smurfed flies (*Figure 8A*). In contrast, Smurfed and non-Smurfed RAL flies with low (RAL774) or high (RAL892) $MI_{24}$ had similar levels of FOXO target gene expression when cultured on molasses food after the primary injury (*Figure 8B,C*), as did Smurfed and non-Smurfed $w^{1118}$ flies when cultured on 1.2 M sucrose after injury (*Figure 8D*). Thus, impaired insulin signaling does not appear to explain the increase in glucose concentration in hemolymph caused by traumatic injury nor the subsequent mortality.

## Discussion

### The HIT device fly model replicates central features of human TBI

TBI in humans can result from a strong mechanical jolt to the body or head that causes the brain to collide against the rigid skull (*Davis, 2000*; *Masel and DeWitt, 2010*). Initial or primary damage from impact forces results in brain dysfunction manifested by a variety of symptoms, including loss of consciousness, seizure, and other behavioral and cognitive impairments. Subsequently, additional non-mechanical secondary injuries can arise over time in response to the primary injuries resulting in further pathological consequences, which in the most severe cases includes death. The biological

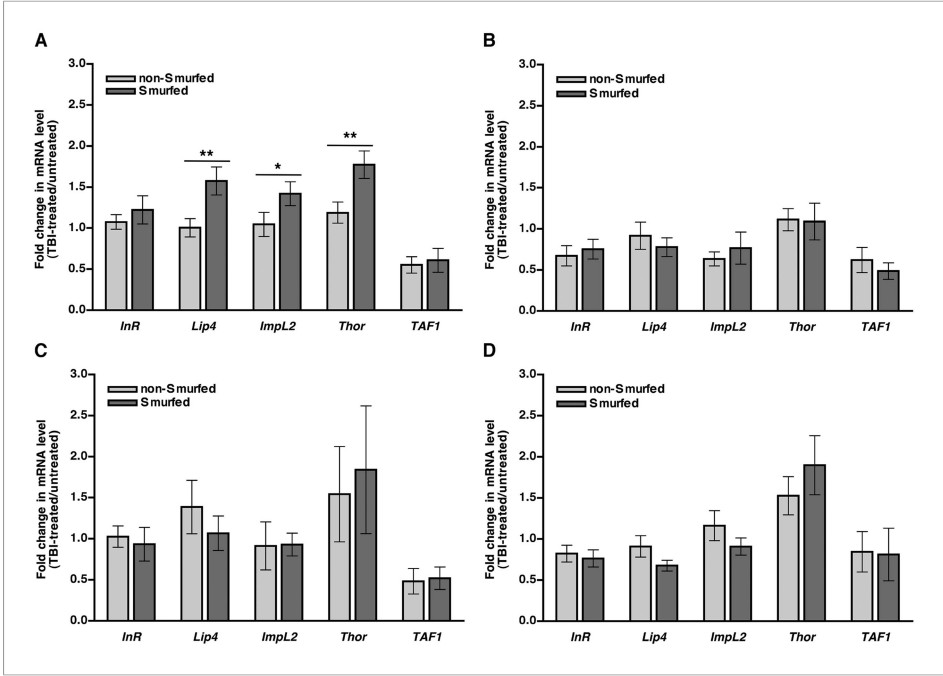

**Figure 8**. In general, FOXO target gene expression is not affected in response to traumatic injury. mRNA expression level of the indicated genes normalized to *actin* expression in non-Smurfed (light gray bars) and Smurfed (dark gray bars) flies 2 hr after the standard injury protocol. (**A**) *w*[1118] flies cultured on molasses food after the primary injury, (**B**) RAL774 flies cultured on molasses food after the primary injury, (**C**) RAL892 flies cultured on molasses food after the primary injury, and (**D**) *w*[1118] flies cultured on 1.2 M sucrose after the primary injury. *InR*, *Lip4*, *Impl2*, and *Thor* are FOXO target genes, and *TAF1* is not a FOXO target gene. *p < 0.05, **p < 0.01, one-tailed *t* test.

pathways linking primary and secondary injuries and the proximate cause of death following TBI are poorly understood at present. We have recently developed a traumatic injury model in *Drosophila* that replicates many of the key features of TBI in humans (*Katzenberger et al., 2013*). Here, we investigate the link between primary and secondary injuries in this model and determine which pathophysiological manifestations correlated with death 24 hr after injury. Our results suggest that subjecting flies to collision impact by the HIT device results in primary TBI that triggers secondary injuries, including damage to the intestinal barrier. The resulting increased intestinal permeability is highly correlated with mortality within 24 hr after injury and is very likely to be one of the main causative factors of death.

Because TBI in humans is diagnosed based on symptoms rather than by any precise medical test, and because flies subjected to mechanical impact injury by our HIT device do not exclusively contact the vial wall with their head, it is difficult to prove rigorously that we have caused a TBI-like injury in flies. Nonetheless, our results are consistent with the supposition that at least some fraction of the flies subjected to the HIT device have suffered an injury analogous with TBI in humans. They exhibit diagnostic features associated with human TBI, notably temporary loss of motor activity with flies lying incapacitated on their back or side, with no evidence of external mechanical damage (*Figure 2—figure supplement 1*). Motor activity recovers gradually over a 5-min period, although ataxia persists for a longer period. In addition to these immediate impairments, over a more extended period, the injured flies also manifest a shortened lifespan, behavioral deficits, and onset of neurodegeneration in the central brain (*Katzenberger et al., 2013*). These phenotypes are all indicative of brain dysfunction, which is the defining characteristic of TBI. Indeed, screens in *Drosophila* for reversible, conditional loss of motor activity, resulted in the isolation of mutants with defects in neuronal excitability and synaptic transmission, which were further enriched for neurodegenerative phenotypes (*Siddiqi and Benzer, 1976*; *Littleton et al., 1998*; *Palladino et al., 2002*, *2003*; *Fergestad et al., 2006*; *Gnerer et al., 2006*; *Babcock et al., 2015*). Activation of the innate immune response in the brain following treatment by the HIT device is a further indication of injury to

the brain (*Katzenberger et al., 2013*). Moreover, because flies become paralyzed immediately after mechanical injury, this must be a primary injury response. The high-speed movie shows that flies subjected to the HIT device could sustain primary injuries to the brain through multiple mechanisms: acceleration-deceleration forces on the brain due to contact of the head with the vial, coup contrecoup forces on the brain due to rebounding of the spring, or sheer forces on the brain due to unrestricted head movements that accompany contact of the body with the vial (*Balsiger et al., 2014*). Thus, although we cannot rule out other more complicated interpretations, as a working hypothesis, we believe our data indicate that some percentage of flies subjected to mechanical impact by the HIT device suffer a brain injury that shares significant features with TBI in humans.

## Death following TBI in flies is associated with intestinal barrier dysfunction

In this study, we focused on investigating the underlying causes of mortality within 24 hr after subjecting flies to TBI. Since flies that die within this period do not do so immediately after the primary injury, there must be some secondary effect that amplifies the initial injury to cause death. Unexpectedly, our results point to intestinal barrier dysfunction as a physiological consequence following TBI that is a major factor in subsequent mortality. Four lines of evidence support this conclusion. (1) GWA analysis for variation in the $MI_{24}$ uncovered genes linked to the function of septate junctions (*Figure 1A* and *Supplementary file 1*), including *grh*, which encodes a transcription factor required for epithelial barrier formation, and *bbg* and *scrib*, which encode PDZ domain-containing, septate junction-associated proteins (*Bilder and Perrimon, 2000*; *Narasimha et al., 2008*; *Bonnay et al., 2013*). (2) There was a very high correlation between the $MI_{24}$ and the onset of Smurfing, a reporter of increased intestinal permeability (*Figure 2C*). (3) There was a strong correlation between the $MI_{24}$ and intestinal leakage of glucose that was ingested after a primary injury (*Figures 4B,C, 7*). (4) There was a significant correlation between activation of the innate immune response by leakage of bacteria from the intestine and the $MI_{24}$ (*Figure 6* and *Figure 6—figure supplements 1–4*).

Our results also provide evidence that intestinal barrier dysfunction is secondary to brain injury inflicted by the HIT device. In particular, leakage of glucose from the intestine, which is delayed relative to the time of injury by the HIT device (*Figure 4B*), and the increased probability of intestinal permeability by ingestion of food after injury (*Figure 5C*) both indicate that intestinal permeability is a secondary response to the initial mechanical injury. Moreover, we found that direct injury to the brain via a crushing injury, is sufficient to trigger intestinal barrier dysfunction (*Figure 2B*). This conclusion is supported by the observation that flies immediately incapacitated following injury by the HIT device, which are the ones most likely to have suffered a brain injury, had a significantly higher probability of death, which is associated with intestinal barrier dysfunction, than non-incapacitated flies (*Figure 2—figure supplement 1*).

Physiological events associated with death following traumatic injury in flies are shared with TBI in mammals. Gastrointestinal dysfunction, including increased intestinal permeability, is frequently observed in TBI patients (*Krakau et al., 2006*). Moreover, increased intestinal permeability occurs in rodent TBI models in which injury is inflicted exclusively to the brain, demonstrating that increased intestinal permeability can result from direct mechanical injury to the brain (*Hang et al., 2003*; *Feighery et al., 2008*; *Jin et al., 2008*; *Bansal et al., 2009*, *2010*). While increased intestinal permeability is linked to death in critically ill patients and correlates with the severity of injury in trauma patients, it has not yet been linked to death in TBI patients (*Doig et al., 1998*; *Faries et al., 1998*; *Reintam et al., 2009*; *Tude Melo et al., 2010*; *Piton et al., 2011*). On the other hand, patients with severe TBI have significantly higher blood glucose levels than patients with moderate or mild TBI, and hyperglycemia is highly predictive of death following TBI (*Rovlias and Kotsou, 2000*; *Salim et al., 2009*; *Tude Melo et al., 2010*; *Harun Harun et al., 2011*; *Prisco et al., 2012*; *Alexiou et al., 2014*; *Elkon et al., 2014*; *Yuan et al., 2014*). In addition, patients with diabetes mellitus, a disease characterized by insulin resistance, have an increased risk of death following TBI (*Ley et al., 2011*; *Lustenberger et al., 2013*). However, thus far, modulating blood glucose levels by intensive insulin treatment in humans has had no effect on the probability of death within 6 months of a primary injury (*Bilotta et al., 2008*; *Yang et al., 2009*). Thus, additional research is still need to understand the mechanistic relationship between blood glucose levels and the probability of death following TBI. Because key features appear to be conserved between flies and mammals, further studies using the fly TBI model should help provide important new information.

Taken together, these data support the conclusion that primary TBI triggers secondary intestinal barrier dysfunction (*Figure 9*). Consequent leakage of sugars across the impaired intestinal barrier causes further impairment of this barrier and ultimately death through an unknown proximal event.

## TBI and aging have similar physiological consequences

Physiological phenotypes of flies that die from TBI are shared with those of flies that die from old age. *Rera et al. (2012)* found that, regardless of chronological age, a few days prior to death, $w^{1118}$ flies show increased intestinal permeability and increased activation of the innate immune response. They also found that $w^{1118}$ flies had reduced insulin signaling, which we observed with $w^{1118}$ flies cultured on molasses food, but, paradoxically, not with other fly lines or under other conditions (*Figure 8*). In addition, we previously found that the probability of death following TBI increases with age in flies, and others have found that the probability of death within one or 6 months of a primary injury increases with age in TBI patients (*Susman et al., 2002*; *Hukkelhoven et al., 2003*; *Dhandapani et al., 2012*; *Katzenberger et al., 2013*). Taken together, these data suggest that as flies and humans age, events such as breakdown of the intestinal epithelial barrier progressively become more severe until they are sufficient to cause death. Thus, as has been suggested for neuropathologies in human TBI (*Smith et al., 2013*), TBI may cause death in young flies by triggering secondary injury mechanisms that parallel the defects that otherwise occur as part of the normal aging process. Older flies would then be more susceptible to death following TBI than younger flies because some critical physiological mechanism such as breakdown of the intestinal epithelial barrier would already be compromised to the point where it would require less of a subsequent insult to push the impairment beyond the threshold for lethality.

## The fly TBI model has provided new insights into the mechanisms underlying death following TBI

Our results demonstrate that the probability of death following TBI is a quantitative trait likely to be affected by many genes. This discovery was possible because we were able to examine many genetically diverse wild-type fly lines under conditions where factors known to affect the probability of death following TBI, that is, the force, number, and timing of primary injuries, age at the time of the primary injury, and environmental conditions, were kept constant (*Figure 1A* and *Figure 1—figure supplement 1*). Furthermore, through GWA analysis, we were able to identify 216 SNPs in 98 genes that are significantly associated with the probability of death following TBI (*Supplementary file 3*). Presumably, these SNPs create a physiological state that alters the severity of primary and/or secondary injuries. For example, SNPs in genes that affect the function of septate junctions (*grh*, *bbg*, and *scrib*) or glucose homeostasis (*sxc*, *pgant2*, *miR-14*, and *CG7882*) may generate intestinal epithelial barriers that are sensitive to disruption by cellular and molecular mechanisms triggered by TBI. Genotype is likely to play an equally important role in humans since variation in several genes has already been shown to be associated with the probability of death of severe TBI patients (*Dardiotis et al., 2010*; *Garringer et al., 2013*; *Failla et al., 2015*). These data suggest that SNPs in orthologs of genes such as *grh* will correlate with the probability of death of TBI patients and could be used for genetic susceptibility testing to identify individuals at high risk of death following TBI.

Nevertheless, it remains to be determined whether any of the 98 genes identified by our GWA study directly affect the probability of death

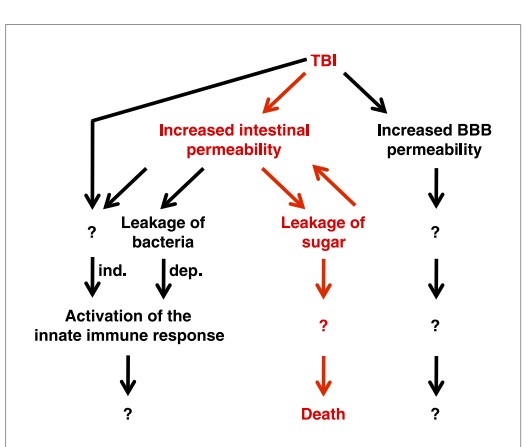

**Figure 9**. Genetic, cellular, and molecular data presented in this study suggest a model for the pathway of events following TBI. Intestinal barrier dysfunction may play an important role in promoting death following TBI (indicated in red). TBI induces additional physiological changes (indicated in black) that do not cause death but may contribute to other outcomes such as neurodegeneration. Bacteria-independent and bacteria-dependent pathways that activate the innate immune response are indicated by 'ind.' and 'dep.', respectively.

following TBI. We attempted to determine this for *grh* by examining the $MI_{24}$ of existing *grh* mutant fly lines. We found that the $MI_{24}$ ranged from $18.9 \pm 2.8$ to $31.6 \pm 0.9$ for 0–7 day old flies that were heterozygous for different *grh* mutant alleles, two lethal P-element alleles and three lethal EMS alleles. However, determining whether and how these *grh* mutations affect the $MI_{24}$ requires comparison with isogenic flies that are wild-type for *grh*, which unfortunately do not exist. Moreover, we do not yet know how the identified SNPs affect *grh* expression or function. For example, if the SNPs increase *grh* expression or alter only a subset of *grh* functions then *grh* loss-of-function mutations would not be expected to have the same effect on the $MI_{24}$. Use of the CRISPR technology in future experiments will help resolve these issues (*Gratz et al., 2013*).

Our studies have also led to the identification of additional physiological events evoked by TBI, including BBB dysfunction (*Figure 3*), leakage of bacteria from the intestinal lumen (*Figures 4A, 5*), and activation of the innate immune response (*Figure 6* and *Figure 6—figure supplements 1–4*). Although occurrence of these events is not correlated with the probability of death following TBI, we hypothesize that they are not benign. One likely possibility is that they are associated with unfavorable long-term TBI outcomes in flies and humans that survive for an extended period after injury. For example, since activation of the innate immune response in the brain causes neurodegeneration in flies and is a common feature of human neurodegenerative diseases, it may be an important factor for neurodegeneration in the fly TBI model and for chronic traumatic encephalopathy, a form of neurodegeneration, in TBI patients (*Tan et al., 2008*; *Arroyo et al., 2011*; *Chinchore et al., 2012*; *Petersen et al., 2012, 2013*; *Cao et al., 2013*; *Baugh et al., 2014*). In support of this idea, the β-lactam antibiotic ceftriaxone has neuroprotective effects in a rat TBI model (*Wei et al., 2012*; *Goodrich et al., 2013*; *Cui et al., 2014*). It will be interesting to determine if reducing the level of activation of the innate immune response by feeding flies antibiotics reduces the severity of neurodegeneration following TBI.

Finally, we found that the amount of sugar ingested immediately after a primary injury greatly affects the probability of death (*Figure 7*). There is considerable evidence that diet after a primary injury influences the outcomes of TBI patients (*Greco and Prins, 2013*; *Scrimgeour and Condlin, 2014*). For example, zinc supplementation reduces the probability of death of severe TBI patients, and the amount of nutrition in the first 5 days after a primary injury affects the probability of death of severe TBI patients (*Young et al., 1996*; *Härtl et al., 2008*). Our results in *Drosophila*, suggest that limiting sugar intake immediately after TBI in humans may be worth investigating as a therapeutic option to reduce the probability of death. Moreover, evolutionary conservation of the intestinal response to TBI between flies and humans suggests that elucidation of the underlying genotype- and age-dependent mechanisms in flies will have clinical relevance.

In summary, these studies have shown that key phenotypic manifestations of TBI and the underlying physiological mechanisms are shared between *Drosophila* and humans. By exploiting the many experimental advantages offered by a *Drosophila* TBI model, it should be possible to obtain novel information to gain further insight into the biology of TBI and ultimately derive new therapeutic strategies to limit its deleterious outcomes in humans.

## Materials and methods

### Fly lines and culturing

Flies were maintained on molasses food at 25°C unless otherwise stated. Molasses food contained 30 g Difco granulated agar (Becton-Dickinson, Sparks, MD), 44 g YSC-1 yeast (Sigma, St. Louis, MO), 328 g cornmeal (Lab Scientific, Highlands, NJ), 400 ml unsulphured Grandma's molasses (Lab Scientific), 3.6 l water, 40 ml propionic acid (Sigma), and tegosept (8 g Methyl 4-hydroxybenzoate in 75 ml of 95% ethanol) (Sigma). Water and sucrose, glucose, and fructose (all from Sigma) vials were prepared immediately before use by placing a circular piece of Whatman filter paper (GE Healthcare Bio-Sciences, Pittsburgh, PA) at the bottom of the vial to absorb 200 µl of liquid. Molasses food with antibiotics contained 100 µg/ml ampicillin (Fisher Scientific, Fair Lawn, NJ), 50 µg/ml vancomycin (Sigma), 100 µg/ml neomycin (Sigma), and 100 µg/ml metronidazole (Sigma) in standard molasses food, as described by *Liu et al. (2012)*. The DGRP collection of flies was obtained from the Bloomington Stock Center, the African collection was provided by John Pool (UW-Madison), and *grh* mutants were provided by Melissa Harrison (UW-Madison) (*Mackay et al., 2012*; *Bastide et al., 2014*).

## Physiological and molecular assays

The MI$_{24}$ was determined as described in *Katzenberger et al. (2013)*. Fly lines not treated with the HIT device had ≤1.15% death within the 24 hr period examined. SNPs associated with the probability of death following TBI were identified using the DGRP Freeze 1 and 2 web tools (*Mackay et al., 2012*; *Huang et al., 2014*). Intestinal permeability was determined using the Smurf assay, as described by *Rera et al. (2011)*, *(2012)*. BEB permeability was determined using the fluorescence assay described by *Pinsonneault et al. (2011)*, except that tetramethylrhodamine-conjugated dextran (Life Technologies, Grand Island, NY) was used as the probe. Hemolymph was extracted from flies by centrifugation, as described by *Tennessen et al. (2014)*, except that flies were decapitated rather than punctured and hemolymph was extracted from heads and bodies. Also, the glass wool was packed tightly to block passage of solid material and the collected hemolymph was thoroughly mixed to resuspend bacterial that may have pelleted during centrifugation. Glucose concentration in hemolymph was performed using the glucose oxidase (GO) assay (Sigma), as described by *Tennessen et al. (2014)*. Bacterial counts in whole flies were performed as described by *Liu et al. (2012)*. Bacterial counts in hemolymph were determined by diluting 1 μl of hemolymph into 50 μl of LB, spreading the whole sample on an LB plate, and counting the number of colonies after 2 day at 25˚C. Quantitative real-time reverse transcription PCR (qRT-PCR) was performed on total RNA extracted from whole flies as described in *Petersen et al. (2012)*. PCR of 16S rDNA was performed using 1 μg of total DNA extracted from flies and primers that amplify 16S rDNA from most eubacteria (*Weisburg et al., 1991*). PCR primer sequences are listed in *Supplementary file 5*.

## Acknowledgements

We thank Grace Boekhoff-Falk and members of the Wassarman and Ganetzky laboratories for their insights that greatly improved this research; Zac Balsiger, Jon Leudtke, Scott Mawer, and Malachi Willey for producing the movie of the HIT device; John Pool, Melissa Harrison, and the Bloomington Stock Center for providing flies; Bernie Weisblum for help with the glucose assay; and Wen Huang for assistance analyzing and interpreting the DGRP data. This work was supported by the National Institutes of Health (R01 AG033620 to BG) and by Robert Draper Technology Innovation Funding (to DAW).

## Additional information

### Funding

| Funder | Grant reference | Author |
|---|---|---|
| National Institutes of Health (NIH) | R01 AG033620 | Barry Ganetzky |
| University of Wisconsin-Madison | Robert Draper Technology Innovation Funding | David A Wassarman |

The funders had no role in study design, data collection and interpretation, or the decision to submit the work for publication.

### Author contributions

RJK, SC, SAR, Conception and design, Acquisition of data, Analysis and interpretation of data, Drafting or revising the article; JAF, GK, JMS, LCS, JEZ, Acquisition of data, Analysis and interpretation of data; BG, DAW, Conception and design, Analysis and interpretation of data, Drafting or revising the article

## Additional files

### Supplementary files

• Supplementary file 1. MI24 values for RAL lines (listed by increasing MI24).

• Supplementary file 2. MI24 of African lines (listed by increasing MI24).

• Supplementary file 3. SNPs associated with the MI24 (listed alphabetically based on gene symbol).

- Supplementary file 4. The occurrence of selected SNPs in RAL lines examined in the figures.

- Supplementary file 5. Primers for PCR.

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
