## [Decision Letter]

Thank you for sending your work entitled “Death following traumatic brain injury in *Drosophila* is associated with intestinal barrier dysfunction” for consideration at *eLife*. Your article has been favorably evaluated by K VijayRaghavan (Senior editor), a Reviewing editor, and 4 reviewers.

The Reviewing editor and the reviewers discussed their comments before we reached this decision, and the Reviewing editor has assembled the following comments to help you prepare a revised submission.

All reviewers praised the creative use of *Drosophila* in studying an important problem and they found the manuscript to be interesting and agreed that it has made valuable research progress. They would be keen to see this manuscript published if a central concern is addressed. Furthermore, there were several technical concerns raised that we have listed below that we hope you can address.

Critical issues:

1) The issue that troubled each reviewer was that they are not at all convinced that the techniques employed for causing the injury results in localized trauma to the brain. Therefore the whole premise that it is brain injury that is causal in this process is not convincing when the entire animal is subjected to trauma; BBB, injury to other body parts, all could be affected. Also, the order of events, well established in the case of mammalian TBI studies is not clear here. The reviewers will like to see some validation that the brain injury is the primary cause. An alternate method that generates the trait is essential to support the final claim. And, if such an alternative method is unable to elicit the stated responses, then perhaps the authors can revise their model.

2) The observed effects in *Drosophila*, if confirmed are interesting enough for the reviewers and likely for the readers of *eLife* as well. Although a fly model of TBI will be nice, it is premature to emphasize this aspect to the extent that has been done in the manuscript.

3) The association with *grh* is interesting and the reviewers think that the paper will be even more interesting if this correlation holds up. We do agree that it is technically challenging to compare mutants with isogenic lines. One approach that could prove effective would be to use an inducible driver system (e.g. Gene-Switch) in combination with adult-specific RNAi of *grh*.

This will make the paper better, but the reviewers agreed that detailed mechanistic studies are outside the scope of this manuscript and would not like to hold up this publication if this is difficult to achieve in a short time-frame.

Technical issues:

1) Hemolymph extraction method. Decapitation and centrifugation as a method of hemolymph extraction is likely to introduce contaminants from the fly's internal tissues into the “hemolymph” samples. Of particular concern is contamination by bacteria and nutrients from the gut. This greatly impacts the author’s claim that increases in both bacterial counts and glucose levels are specific to the hemolymph. Two suggestions: the authors could use the protocol detailed here, http://www.jove.com/video/2722/insulin-injection-hemolymph-extraction-to-measure-insulin-sensitivity, or make careful use of a capillary needle, to extract hemolymph without disrupting the gut. Glucose homeostasis can change very rapidly, and glucose release from other damaged tissues, or a reduction in glucose uptake as a result of whole-fly trauma, cannot be ruled out. This latter part can be addressed by careful rewriting, but an alternative to the centrifugation method is essential at least as a control.

2) Contamination from the fly's surface bacteria. The authors state that traumatic injury results in a less mobile animal, at least for a time following injury and may well spend longer than usual in contact with bacteria. The flies should be surface sterilized with ethanol wash prior to measuring bacterial load differences. Also, treatment with antibiotic and colony growth does not ensure that bacteria that do not grow well on regular plates are not present. The authors should avail themselves of proper published axenic assays.

3) It is possible that instead of the sugar being toxic, the 'water diet' is protective. This could be discussed as an alternative.

4) An issue with fly GWAS analysis is that 2.5 million SNPs are tested, yet sites with P-values below 10^-5 are considered as significant associations. Bonferroni correction for multiple tests would be on the order of 10^-8. An FDR would make more sense, but it would almost certainly be lower than 10^-7. Thus, many of the “associations” described likely do not survive a reasonable correction for multiple tests. Furthermore, since the panel is small, and there is no attempt to replicate the association at the *grh* gene (using either other quantitative genetics or some functional assay), it is not unlikely the association is simply wrong. The authors should add this word of caution to their analysis.

5) The authors may wish to modify several speculative comments in the manuscript, for example, in the Abstract: “Our results indicate that natural variation in the probability of death following TBI is largely due to genetic differences that affect intestinal barrier dysfunction.” While a genetic component to TBI-related death is very likely, the heritability is not calculated it is not possible to estimate the fraction of the phenotypic variation due to genetic factors. In addition, no evidence is provided that the best GWAS candidate (*grh*) directly impacts the intestinal barrier.

6) It is surprising that there is very little analysis of brain structure. Comparing brain structural integrity in injured flies, and especially after ingestion of water or glucose post injury, will test whether reduced glucose intake after TBI may help preserve brain integrity.

7) In the data on the effect of sugars show that the effective concentrations of the individual components in molasses are higher than they are in molasses. This begs the question how the effects in molasses can be explained (synergistic effects or other alternatives)?

[Editors' note: further revisions were requested prior to acceptance, as described below.]

Thank you for resubmitting your work entitled “Death following traumatic brain injury in *Drosophila* is associated with intestinal barrier dysfunction” for further consideration at *eLife*. Your revised article has been evaluated by K VijayRaghavan (Senior editor), a member of the Board of Reviewing Editors, and also by the original external reviewers of the manuscript. As you know, the most important concern expressed by all reviewers was whether the injuries can be caused in a brain specific way in order to test the central thesis of the manuscript. With the additional “forceps squeezing” method included, the revised version is potentially much improved. Unfortunately, the data on this central point are not definitive.

The picture of a single fly that has had its head squeezed while being fed blue dye is shown. This fly is now a “Smurf”. However, we are left guessing with regards to many basic questions about this phenomenon/experiment: How many flies were tested? What fraction of flies ‘Smurfed’ after the head squeezing? All of them? How many flies died? These are very basic questions and, hopefully, this information is at hand without the need for further work. But such statistics are essential and should support the central thesis in a way comparable to that obtained from the HIT experiments for this paper to be accepted.

Please address the following two issues by Reviewer 2, by rewriting the corresponding sections acknowledging the potential drawbacks a bit more descriptively and clearly in the manuscript.

Reviewer #2

1) One of my other concerns from the previous review was about how the GWAS was carried out. The response to reviews has me even more concerned. The authors previously used the DGRP1 analysis pipeline from the Mackay lab, finding 98 genes with associations, including SNPs at *grh*: one of the candidate genes that led them down the path of considering intestinal barrier problems as a cause of death in their TBI model, and a gene discussed at length in the manuscript. In the response, but critically not in the manuscript, they acknowledge that a re-analysis of the GWAS with the DGRP2 analysis pipeline (which accounts for relatedness, along with a couple of other changes that should make associations more robust) finds only 10 of the previous 98 genes with associations. The set of 10 does not include *grh*. There simply must be a detailed discussion in the paper of the differences between the results with the two pipelines. Since the previous list of 98 genes included several (*grh*, *bbg*, and *scrib, and sxc*, *pgant2*, *miR-14*, and *CG7882*) with a priori interesting functional roles, it is also critical to specifically state whether these genes were replicated in the new pipeline. I appreciate that the GWAS was basically used as a guide to the kinds of pathways that are involved in the trait, but if few of the genes are replicated between pipelines that is a problem for the paper and for the argument.

2) I still don't understand what the problem is with doing a *grh* RNAi. You can purchase the background stock that the TRiP RNAi constructs where injected into, so controlling genetic background is facile.

---

## [Author Response]

*1) The issue that troubled each reviewer was that they are not at all convinced that the techniques employed for causing the injury results in localized trauma to the brain. Therefore the whole premise that it is brain injury that is causal in this process is not convincing when the entire animal is subjected to trauma; BBB, injury to other body parts, all could be affected. Also, the order of events, well established in the case of mammalian TBI studies is not clear here. The reviewers will like to see some validation that the brain injury is the primary cause. An alternate method that generates the trait is essential to support the final claim. And, if such an alternative method is unable to elicit the stated responses, then perhaps the authors can revise their model*.

We fully appreciate this concern and have made considerable revisions to the manuscript to address it. We believe that our new data support the conclusion that flies subjected to mechanical impact by the HIT device suffer a primary brain injury that triggers secondary intestinal barrier dysfunction, which is major factor in subsequent mortality. In addition, our new data provide additional evidence that the physiological consequences of HIT device treatment of flies are similar to those of TBI in humans. First, we have included new data indicating that injury exclusively to the fly brain is sufficient to increase intestinal permeability (Figure 2). Specifically, we found that compression of the head using forceps causes Smurfing (a reporter of intestinal permeability) within 24 hours. Communication between the injured brain and the intestine is conserved in mammals. Several groups have shown using rodent TBI models that direct injury to the brain causes increased intestinal permeability. Thus, these data indicate that TBI induced by the HIT device could cause secondary injures to the intestine that increase barrier permeability and that this finding is relevant to mammals. Second, our new data on incapacitated flies (Figure 2—figure supplement 1) indicate that flies subjected to the HIT device receive TBI and die from these injuries. Incapacitation in flies is similar to the symptoms of a concussion in humans. It is also similar to temporary paralysis observed in bang-sensitive *Drosophila* mutants following mild mechanical shock that disrupts normal electrical activity in the brain. These observations are consistent with the idea that at least a fraction of the flies subjected to the HIT device suffer a brain injury that temporarily disturbs normal neuronal function resulting in temporary paralysis. Furthermore, we found that incapacitated flies had a significantly higher probability of death following the primary injury than non-incapacitated flies, indicating that the injuries to the brain that cause incapacitation also contribute significantly to mortality within 24 hours. Third, we have included new data that indicate that intestinal dysfunction is a secondary injury (Figure 4). Figure 4 shows that the HIT device-induced increase in glucose level in the hemolymph does not occur until 1-2 hours after the primary injury from the HIT device, suggesting that increased intestinal permeability is not a direct consequence of mechanical damage from the HIT device but rather a secondary consequence of molecular and cellular events evoked by the mechanical damage. In addition, Figure 4, which analyzes hemolymph glucose levels in flies fed water rather than molasses food after the primary injury, shows that the increase in glucose levels is due to food ingested *after* the primary injury, further supporting the conclusion that intestinal dysfunction is a secondary injury. These data, taken together with data in the original manuscript, support the conclusion that primary TBI triggers secondary intestinal barrier dysfunction. Consequent leakage of sugars across the impaired intestinal barrier causes further impairment of this barrier and ultimately death through an unknown proximal event.

Because TBI in humans is diagnosed based on symptoms rather than by any precise medical test, and because flies subjected to mechanical impact injury by our HIT device do not exclusively contact the vial wall with their head, it is difficult to prove rigorously that we have caused a TBI-like injury in flies. Nonetheless, our results are consistent with the supposition that at least some fraction of the flies subjected to the HIT device have suffered an injury analogous with TBI in humans. In addition, the fact that primary injuries inflicted by the HIT device are not limited to the brain can be viewed as an advantage of the system because polytrauma (traumatic injury to multiple body parts) often accompanies TBI and is thought to modulate TBI phenotypes. In the revised Discussion section, we have added two sections at the beginning that lay out all of the evidence that brain injury is causal for mortality following treatment with the HIT device. We have also made revisions to the Results section that address how individual pieces of data impact the claim that the primary injuries to the brain cause death in the fly model.

*2) The observed effects in* Drosophila*, if confirmed are interesting enough for the reviewers and likely for the readers of* eLife *as well. Although a fly model of TBI will be nice, it is premature to emphasize this aspect to the extent that has been done in the manuscript*.

This is an excellent point that we have addressed by making it clear that injury from the HIT device may affect fly body parts other than the brain. Beginning in the Introduction section, we make it clear that flies treated with the HIT device are likely to receive injury to multiple body parts and, thus, a goal of the study was to identify the injured body part or parts that cause death within 24 hours. Consequently, throughout the Results section we now refer to the primary injury from the HIT device as “traumatic injury” rather than TBI. However, in the Discussion we provide a careful justification for why brain injury (TBI) is emphasized in the Abstract and Title. This justification is partially based on the new data described in response to the first critical issue but also to data in the original submission as well as data in our previous publication (52).

*3) The association with* grh *is interesting and the reviewers think that the paper will be even more interesting if this correlation holds up. We do agree that it is technically challenging to compare mutants with isogenic lines. One approach that could prove effective would be to use an inducible driver system (e.g. Gene-Switch) in combination with adult-specific RNAi of* grh.

*This will make the paper better, but the reviewers agreed that detailed mechanistic studies are outside the scope of this manuscript and would not like to hold up this publication if this is difficult to achieve in a short time-frame*.

We completely agree with the editors and reviewers on this point and we are indeed engaged in pursuing these studies. Unfortunately, they are difficult experiments that require a considerable amount of time to create the proper constructs and generate the necessary strains, which will take much longer to complete. We very much appreciate that the editors and reviewers recognize this and have made allowances for it.

Technical issues:

*1) Hemolymph extraction method. Decapitation and centrifugation as a method of hemolymph extraction is likely to introduce contaminants from the fly's internal tissues into the* “*hemolymph*” *samples. Of particular concern is contamination by bacteria and nutrients from the gut. This greatly impacts the author’s claim that increases in both bacterial counts and glucose levels are specific to the hemolymph. Two suggestions: the authors could use the protocol detailed here,*
http://www.jove.com/video/2722/insulin-injection-hemolymph-extraction-to-measure-insulin-sensitivity*, or make careful use of a capillary needle, to extract hemolymph without disrupting the gut. Glucose homeostasis can change very rapidly, and glucose release from other damaged tissues, or a reduction in glucose uptake as a result of whole-fly trauma, cannot be ruled out. This latter part can be addressed by careful rewriting, but an alternative to the centrifugation method is essential at least as a control*.

Although we understand these concerns, we feel that the controls we have performed provide strong evidence for the adequacy of our method. In both the bacteria and glucose experiments, we have controls that address contamination from the gut. There were no bacteria in the hemolymph of untreated flies, indicating that the gut did not contaminate the hemolymph (Figure 4). The concentration of glucose in the hemolymph of untreated flies was very similar to the concentration determined using other methods for hemolymph extraction (Figure 4) (91). Lastly, if contamination did occur, it would have occurred equally under control and experimental conditions, and our conclusions are based on differences between control and experimental conditions.

*2) Contamination from the fly's surface bacteria. The authors state that traumatic injury results in a less mobile animal, at least for a time following injury and may well spend longer than usual in contact with bacteria. The flies should be surface sterilized with ethanol wash prior to measuring bacterial load differences. Also, treatment with antibiotic and colony growth does not ensure that bacteria that do not grow well on regular plates are not present. The authors should avail themselves of proper published axenic assays*.

Again, although we understand this concern, we feel it is unlikely to be an issue here because bacterial load, as assayed either by plating or by PCR of 16S rDNA, was below detection, so bacteria on the surface of flies is not an issue. Furthermore, we have used a primer set for PCR detection of 16S rDNA that is described “Universal PCR primers for bacteria”, and detects most bacterial species (97).

*3) It is possible that instead of the sugar being toxic, the 'water diet' is protective. This could be discussed as an alternative*.

At present we cannot distinguish between these possibilities so we have added the following sentence to the Results: “However, the present data do not rule out the possibility that water provides protection against death following traumatic injury.”

*4) An issue with fly GWAS analysis is that 2.5 million SNPs are tested, yet sites with P-values below 10^-5 are considered as significant associations. Bonferroni correction for multiple tests would be on the order of 10^-8. An FDR would make more sense, but it would almost certainly be lower than 10^-7. Thus, many of the* “*associations*” *described likely do not survive a reasonable correction for multiple tests. Furthermore, since the panel is small, and there is no attempt to replicate the association at the* grh *gene (using either other quantitative genetics or some functional assay), it is not unlikely the association is simply wrong. The authors should add this word of caution to their analysis*.

We thank the reviewers for providing this cautionary note. We have tried to acknowledge it by adding the following sentence to the Results section: “However, despite the small *P*-values, some of the associations may be false positives because the minor allele frequency cut-off of the DGRP Freeze 1 algorithm was 4 lines, allowing the *P*-value to be driven by a few extreme lines”.

*5) The authors may wish to modify several speculative comments in the manuscript, for example, in the Abstract:* “*Our results indicate that natural variation in the probability of death following TBI is largely due to genetic differences that affect intestinal barrier dysfunction.*” *While a genetic component to TBI-related death is very likely, the heritability is not calculated it is not possible to estimate the fraction of the phenotypic variation due to genetic factors. In addition, no evidence is provided that the best GWAS candidate (*grh*) directly impacts the intestinal barrier*.

We have modified the Abstract to say: “Our results indicate that natural variation in the probability of death following TBI is due in part to genetic differences that affect intestinal barrier dysfunction”.

*6) It is surprising that there is very little analysis of brain structure. Comparing brain structural integrity in injured flies, and especially after ingestion of water or glucose post injury, will test whether reduced glucose intake after TBI may help preserve brain integrity*.

In our earlier publication (52), we performed brain histology and found that prior to the onset of neurodegeneration at a later age, we do not see any overt damage to the brain injured flies. Furthermore, we do not expect to see major structural defects vs. functional defects from disturbances in brain activity. We think that electrophysiological assays are likely to be more informative but are beyond the scope of the present investigation.

*7) In the data on the effect of sugars show that the effective concentrations of the individual components in molasses are higher than they are in molasses. This begs the question how the effects in molasses can be explained (synergistic effects or other alternatives)*?

The unknown variable in this assay is the amount of food that the flies ingest. It is possible that the effective concentration of individual sugars is higher than in molasses because flies ingest a smaller volume of the individual sugars. Regardless, the point made by the reviewer is valid, so we have added the following sentence to the Results section: “The effective concentrations of the individual sugars are higher than they are in molasses suggesting that sugars in combination have a synergistic effect on the MI_24_ or that another component of molasses food is important.”

*[Editors' note: further revisions were requested prior to acceptance, as described below*.*]*

*As you know, the most important concern expressed by all reviewers was whether the injuries can be caused in a brain specific way in order to test the central thesis of the manuscript. With the additional* “*forceps squeezing*” *method included, the revised version is potentially much improved. Unfortunately, the data on this central point are not definitive*.

*The picture of a single fly that has had its head squeezed while being fed blue dye is shown. This fly is now a* “*Smurf*”*. However, we are left guessing with regards to many basic questions about this phenomenon/experiment: How many flies were tested? What fraction of flies ‘Smurfed’ after the head squeezing? All of them? How many flies died? These are very basic questions and, hopefully, this information is at hand without the need for further work. But such statistics are essential and should support the central thesis in a way comparable to that obtained from the HIT experiments for this paper to be accepted*.

As suggested, we have quantified the head compression data. These data support the conclusion that brain injury can cause Smurfing and death. These results are reported in the subsection “Traumatic injury causes intestinal barrier dysfunction” of the manuscript as follows: “We found that brain injury caused by compressing the head of 0-7 day old *w*^*1118*^ flies from eye-to-eye using forceps was sufficient to cause Smurfing within 24 hours (Figure 2). […] increased intestinal permeability of flies subjected to the HIT device is due to brain injury.”

*Please address the following two issues by Reviewer 2, by rewriting the corresponding sections acknowledging the potential drawbacks a bit more descriptively and clearly in the manuscript*.

Reviewer #2

*1) One of my other concerns from the previous review was about how the GWAS was carried out. The response to reviews has me even more concerned. The authors previously used the DGRP1 analysis pipeline from the Mackay lab, finding 98 genes with associations, including SNPs at* grh*: one of the candidate genes that led them down the path of considering intestinal barrier problems as a cause of death in their TBI model, and a gene discussed at length in the manuscript. In the response, but critically not in the manuscript, they acknowledge that a re-analysis of the GWAS with the DGRP2 analysis pipeline (which accounts for relatedness, along with a couple of other changes that should make associations more robust) finds only 10 of the previous 98 genes with associations. The set of 10 does not include* grh*. There simply must be a detailed discussion in the paper of the differences between the results with the two pipelines. Since the previous list of 98 genes included several (*grh*,* bbg*, and* scrib*, and* sxc*,* pgant2*,* miR-14*, and* CG7882*) with a priori interesting functional roles, it is also critical to specifically state whether these genes were replicated in the new pipeline. I appreciate that the GWAS was basically used as a guide to the kinds of pathways that are involved in the trait, but if few of the genes are replicated between pipelines that is a problem for the paper and for the argument.*

As suggested by the reviewer, in the paper, we have expanded upon the discussion of the difference between the results obtained with the DGRP Freeze 1 and 2 algorithms. The Discussion in the manuscript now reads: “This analysis revealed that 216 unique SNPs located in or near 98 genes were associated with the MI_24_ at a discovery significance threshold of *P*<10^-5^ ([Supplementary-material SD3-data]). […] the SNP that was most significantly associated with the MI_24_ (*P*=1.15X10^-10^) as well as three other significant SNPs.”

*2) I still don't understand what the problem is with doing a* grh *RNAi. You can purchase the background stock that the TRiP RNAi constructs where injected into, so controlling genetic background is facile.*

We agree with the reviewer that it important to address the causal relationship between the identified genes and the MI_24_ and that RNAi is a good approach to achieve this goal. However, we feel that these studies are not necessary to support the conclusions in the paper. Additionally, the proposed experiment is more involved than portrayed because the genetic background of driver lines is also an issue, multiple *grh* RNAi lines as well as negative and positive control lines will need to be tested, it is unclear which driver line to use because we do not know where or when *grh* function might be required to affect intestinal permeability, and we do not know whether the *grh* variants in the RAL lines are loss-of-function or gain-of-function. Thus, we think that it would be best to include genetic analyses of *grh* and other genes identified by GWAS in a future manuscript.